# GENERATE, ANNOTATE, AND LEARN: GENERATIVE MODELS ADVANCE SELF-TRAINING AND KNOWLEDGE DISTILLATION

## ABSTRACT

Semi-Supervised Learning (SSL) has seen success in many application domains, but this success often relies on the availability of task-specific unlabeled data. Knowledge distillation (KD) has enabled compressing deep networks, achieving the best results when distilling knowledge on fresh task-specific unlabeled data. However, task-specific unlabeled data can be challenging to find, especially for NLP. We present a simple framework called "generate, annotate, and learn (GAL)" that uses unconditional language models to synthesize in-domain unlabeled data, helping advance SSL and KD on NLP and tabular tasks. To obtain strong task-specific generative models, we either fine-tune a large language model (LLM) on inputs from specific tasks, or prompt a LLM with a few input examples to generate more unlabeled examples. Then, we use existing classifiers to annotate generated unlabeled examples with pseudo labels, which are used as additional training data or as additional prompts. GAL improves prompt-based few-shot learning on several NLP tasks. It also yields a new state-of-the-art for 6-layer transformers on the GLUE leaderboard. Finally, self-training with GAL offers large gains on four tabular tasks from the UCI repository.

## 1 INTRODUCTION

Unlabeled data is abundant in the real world, but task-specific unlabeled data within the scope of a given machine learning problem can be challenging to find. For instance, one cannot easily find in-domain unlabeled data conforming to the input distribution of a specific Natural Language Processing (NLP) task from the GLUE benchmark (Wang et al., 2019b). Some NLP tasks require an input comprising a pair of sentences with a particular relationship between them. Moreover, classification datasets typically represent a tailored distribution of text and only include a limited number of class labels. If task-specific unlabeled data were available, one could adopt self-training (Yarowsky, 1995) to automatically annotate unlabeled data with pseudo labels to improve accuracy and robustness of classifiers (Xie et al., 2020; Carmon et al., 2019b). In addition, one can use knowledge distillation (Hinton et al., 2015) on fresh task-specific unlabeled data to more effectively compress deep neural networks and ensembles (Buciluǎ et al., 2006; Chen et al., 2020c).

When task-specific unlabeled examples do not exist, one can try to retrieve them from a large and diverse open-domain dataset. For instance, Du et al. (2020) have used nearest neighbor retrieval to harvest in-domain unlabeled text from the internet, leading to a successful application of self-training and knowledge distillation to certain NLP tasks. While retrieval can indeed help to find in-domain data for problems with simple inputs, it is not practical for problems with complex input schemes, *e.g.,* sentence pairs with certain relations and tabular data. Accordingly, self-training and retrieval-based methods have not been widely adopted for NLP tasks, *e.g.,* on the GLUE benchmark.

This paper presents a deceptively simple and general framework called "generate, annotate, and learn (GAL)" to help advance semi-supervised learning and knowledge distillation on various applications that do not come with unlabeled data. We advocate for the use of language models to synthesize unlabeled tasks-specific data, in lieu of real unlabeled data. We build on recent advances in text generation (Radford et al., 2019; Gao et al., 2021), and use powerful generative models to synthesize unlabeled text and tables. Then, we use state-of-the-art classifiers to annotate generated unlabeled data with pseudo labels. Finally, we combine labeled data with pseudo labeled data to train more effective classifiers or for the purpose of knowledge distillation (KD).

We motivate GAL by making connections to empirical and vicinal risk minimization (Vapnik, 1992; Chapelle et al., 2001), and demonstrate its utility by presenting empirical results on a wide range of applications. Our key contributions include:

- We propose a simple way to advance SSL, KD, and few-shot learning on NLP by using language models to synthesize large amounts of task-specific unlabeled data.
- We link GAL to empirical and vicinal risk minimization, helping explain why GAL works and why synthetic samples from class-conditional language models are not as effective.
- We systematically dissect GAL and study the key components leading to its success.
- GAL establishes a new SoTA for a single 6-layer transformer on the GLUE test set.
- GAL improves prompt-based few-shot learning, providing an average improvement of 1.3% on four 4-shot learning NLP tasks.
- GAL advance self-training for tabular tasks, outperforming XGBoost on 2 out of 4 tasks.

## 2   RELATED WORK

There has been a surge of interest in improving accuracy and label efficiency of classifiers via:

1. *Self-Supervised pretraining* on open-domain unlabeled data in a task-agnostic way (Peters et al., 2018; Devlin et al., 2019; Chen et al., 2020b),
2. *Self-Training* using domain-specific unlabeled data in a task-specific way (Rosenberg et al., 2005; McClosky et al., 2006; Xie et al., 2020).

While self-supervised learning can be applied to a broad distribution of unlabeled data, self-training requires unlabeled data that at least can be annotated using the same set of class labels available for the downstream task (Oliver et al., 2018). For instance, if one is interested in training a classifier to distinguish images of cats and dogs, self-training with images of aircraft is likely not helpful, but it is conceivable that self-supervised learning with images of aircraft can still help. A growing body of recent work suggests that perhaps self-supervised pretraining and self-training are compatible and can be combined to achieve the best semi-supervised learning performance (Chen et al., 2020c; Du et al., 2020). We corroborate the existing evidence by showing gains from *generative* self-training.

Semi-supervised learning (SSL) has a long and rich history in machine learning (Cooper & Freeman, 1970; McLachlan & Ganesalingam, 1982; Riloff, 1996; Chapelle et al., 2009; Van Engelen & Hoos, 2020). One of the oldest family of SSL algorithms is *self-training*, *a.k.a.* self-learning or self-labeling (Scudder, 1965; Fralick, 1967; Agrawala, 1970; Yarowsky, 1995). Self-training encourages knowledge transfer between a *teacher* and a *student* model in such a way that the student can outperform the teacher. Specifically, one leverages the teacher's knowledge to annotate unlabeled data with so-called *pseudo labels*, and the student learns from a mixture of pseudo- and human-labeled data. Self-training has recently seen renewed interest across vision and NLP applications (Yalniz et al., 2019; Xie et al., 2020; Zoph et al., 2020; Du et al., 2020). Recent work aims to combine self-training and *consistency regularization* to develop powerful SSL algorithms. The key idea is to ensure that the predictions of a classifier on unlabeled examples are robust to strong augmentations (Berthelot et al., 2019a; Sohn et al., 2020; Xie et al., 2019). We build on prior work and investigate the use of synthetic data within the broad family of self-training methods.

Recent theoretical work analyzes self-training for linear models, often under the assumption that the data distribution is (nearly) Gaussian (Carmon et al., 2019a; Raghunathan et al., 2020; Chen et al., 2020d; Kumar et al., 2020a; Oymak & Gulcu, 2020). Wei et al. (2021) prove that, under "expansion" and "class separation" assumptions, self-training can lead to more accurate neural network classifiers. We present a theoretical framing of GAL in terms of empirical and vicinal risk minimization (Vapnik, 1992; Chapelle et al., 2001).

An important family of related work uses generative models for SSL by learning features that are useful for both generation and discrimination (*e.g.,* Chen et al., 2020a; Odena, 2016; Dai et al., 2017). For instance, Kingma et al. (2014) approach SSL by viewing missing class labels as a set of latent variables and use variational inference to impute missing labels as well as other factors of variation. By contrast, our work does not learn features using generative models and keeps the generative and discriminative processes separate. This offers more flexibility and allows GAL to use self-supervised pretraining methods that are not fully generative.

Our work is closely related to recent work on the uses of generative models for data augmentation (Norouzi et al., 2020; Yang et al., 2020). Unlike Norouzi et al. (2020), we do not use instance-

based generative models. Yang et al. (2020) propose a complex scheme, including data relabeling, data filtering, and two-stage training, to utilize synthetic data. By contrast, we show that a simple mixture of the original data and synthetic data can provide sizable gains. Furthermore, we show a more broad use of generative models on KD and few-shot learning, in addition to tabular tasks.

Knowledge Distillation (KD) (Buciluă et al., 2006; Hinton et al., 2015) uses a procedure similar to self-training to distill knowledge of an expressive teacher model into a smaller student model. In contrast, self-distillation (Furlanello et al., 2018; Zhang et al., 2019a; Mobahi et al., 2020) uses teacher and student models of equal size, hoping to iteratively refine class labels. Previous work uses unlabeled data (Buciluă et al., 2006) and adversarial training (Wang et al., 2018) to improve KD. We demonstrate that synthetic data generated by unconditional generative models can improve KD on NLP, outperforming strong KD baselines, which often add more complexity and additional hyper-parameters (*e.g.,* Sun et al., 2019a; Jiao et al., 2019; Xu et al., 2020; Rashid et al., 2021).

Advanced generative models are able to generate realistic images and text (Karras et al., 2017; Brock et al., 2019; Karras et al., 2019; Radford et al., 2019; Brown et al., 2020). The quality of synthetic samples has improved to the extent that deep fake detection has become an important research topic itself (Zellers et al., 2019; Dolhansky et al., 2019). Recent work has aimed to utilize class-conditional generative models to help improve supervised learning (Antoniou et al., 2017; Bowles et al., 2018; Zhang et al., 2019b; Kumar et al., 2020b; Gao et al., 2020). However, Ravuri & Vinyals (2019) have shown that images generated by state-of-the-art class-conditional generative models fall short of improving ImageNet classification accuracy, despite strong sample quality scores (Salimans et al., 2016; Heusel et al., 2017). Similarly, Kumar et al. (2020b) find that it is difficult for sentences generated by label-conditioned GPT-2 (Radford et al., 2019) to retain the semantics or pragmatics of a specified category, which leads to poor performance on downstream tasks. We discuss why class-conditional generative models are hardly effective for supervised learning, and instead, focus on unconditional generative models.

## 3 BACKGROUND ON SELF-TRAINING

Given a labeled dataset $L = \{(\boldsymbol{x}_i, y_i)\}_{i=1}^N$ and an unlabeled dataset $U = \{\boldsymbol{x}_j\}_{j=1}^M$, we summarize the general family of SSL algorithms known as self-training as:

1. First, an initial model denoted $f_1$ is trained using supervised learning on the labeled dataset $L$.
2. Then, at iteration $t$, one adopts $f_t$ as the teacher model to annotate the unlabeled dataset $U$ using *pseudo labels*. Optionally, one uses a selection method to pick a subset $S_t \subseteq \{(\boldsymbol{x}_j, f_t(\boldsymbol{x}_j))\}_{j=1}^M$ of pseudo labeled examples.
3. A student model $f_{t+1}$ is trained to optimize a classification loss on the combination of $L$ and $S_t$:

$$\ell_{t+1} = \mathbb{E}_{(\boldsymbol{x},y)\sim(L \cup S_t)} H(y, f_{t+1}(\boldsymbol{x})) \,, \tag{1}$$

   where $H(q, p) = q^\top \log p$ is the softmax cross entropy loss, and $y$ is assumed to be a one-hot vector (original labels) or a vector of class probabilities (*soft* pseudo labels).
4. Self-training iterations are repeated $T$ times or until performance plateaus.

Many different variants of the basic self-training algorithm discussed above exist in the literature. These variants differ in the type of pseudo labels used, the selection strategy to filter pseudo labeled examples, the speed at which $f_t$ is replaced with $f_{t+1}$, the choice of data augmentation strategy in the teacher and student models, and the weighting of the two datasets in the objective (Berthelot et al., 2019b;a; Xie et al., 2020; Sohn et al., 2020; Du et al., 2020).

An important design choice is the type of pseudo labels used. One can simply use soft class probabilities predicted by a teacher $f_t$ (Du et al., 2020), sharpened class probabilities (Berthelot et al., 2019b), or hard labels (a one-hot vector that is zero except at $\operatorname{argmax} f_t(\boldsymbol{x})$) (Lee et al., 2013). Another important consideration is the selection strategy to retain a subset of pseudo-labeled examples. FixMatch (Sohn et al., 2020) uses a hyper-parameter $\tau$ to select examples on which the teacher model has a certain level of confidence, *i.e.,*

$$S_t = \{(\boldsymbol{x}, f_t(\boldsymbol{x})) \mid \boldsymbol{x} \in U \ \& \ \max(f_t(\boldsymbol{x})) \geq \tau\} \,. \tag{2}$$

NoisyStudent (Xie et al., 2020) also uses a form of confidence filtering but ensures that the class labels in the selected subset are balanced. In principle, any method for out-of-distribution detection (Hendrycks & Gimpel, 2016) can be adopted for filtering pseudo-labeled examples. We adopt the simplest variant of self-training and limit hyper-parameter tuning to a bare minimum.

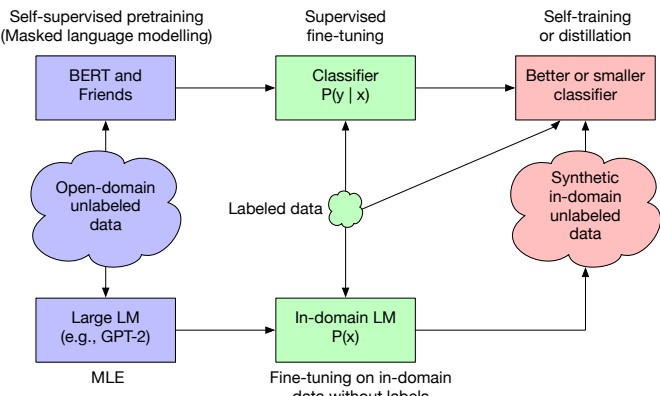

**Figure 1:** An illustration of GAL for NLP. We use open-domain data once for self-supervised pretraining (*e.g.,* BERT) and once for training a large LM (*e.g.,* GPT-2). BERT is fine-tuned on labeled data to yield a classifier for the task of interest. GPT-2 is fine-tuned on the same data without labels to obtain an unconditional task-specific LM, which is used to generate lots of synthetic in-domain unlabeled data for self-training and KD.

## 4 GENERATE, ANNOTATE, AND LEARN (GAL)

Given a labeled dataset $L = \{(\boldsymbol{x}_i, y_i)\}_{i=1}^{N}$, we first train an unconditional domain-specific generative model $g(\boldsymbol{x})$ on $L_x = \{\boldsymbol{x}_i\}_{i=1}^{N}$, and then use it to synthesize unlabeled data. Such synthetic unlabeled data is used within self-training and KD even in the absence of in-domain unlabeled data. We restrict our attention to basic KD and self-training methods, even though GAL can be combined with more sophisticated semi-supervised techniques too.

The objective function of GAL for self-training during iteration $t$, provided a teacher model $f_t$, is expressed as:

$$\ell_{t+1} = \lambda \, \mathbb{E}_{(\boldsymbol{x},y)\sim L} H(y, f_{t+1}(\boldsymbol{x})) + (1-\lambda) \, \mathbb{E}_{\widetilde{\boldsymbol{x}}\sim g(\boldsymbol{x})} H(f_t(\widetilde{\boldsymbol{x}}), f_{t+1}(\widetilde{\boldsymbol{x}})) \,. \tag{3}$$

We use soft pseudo labels within self-training and KD. To improve computational efficiency of GAL, we do not generate unlabeled data on the fly, but generate as many unconditional samples as possible and store them in a synthetic unlabeled dataset $U$. We simply set $\lambda = 0.5$, unless stated otherwise.

Not surprisingly, the effectiveness of GAL depends on the fidelity and diversity of synthetic examples. If we had access to the oracle generative process, we were able to obtain the best KD and SSL results, as if we had access to real task-specific unlabeled data. Our preliminary experiments suggest that large language models are particularly effective within the GAL framework. Hence, as shown in Figure 1, to build the best domain-specific language model, we adopt a large large language model pretrained on lots of open-domain text, and fine-tune it on a given dataset's inputs, *i.e., $L_x$, ignoring class labels*. Both our theory and ablations confirm that ignoring class labels is a good idea. Transferring the knowledge of large language models is particularly beneficial a small input dataset $L_x$ of text is available (Hernandez et al., 2021). In what follows, we first discuss practical considerations around building domain-specific language models, and then link GAL to empirical and vicinal risk minimization, to motivate the approach.

### 4.1 DOMAIN-SPECIFIC GENERATIVE MODELS OF TEXT AND TABLES

**Text.** Many NLP tasks have a relatively small labeled dataset (Wang et al., 2019b;a). While self-supervised pretraining, followed by supervised fine-tuning (Devlin et al., 2019; Liu et al., 2019b; Clark et al., 2020; Lewis et al., 2019) has become the prominent approach to NLP, previous work has also investigated different data augmentation methods to increase the size of the training datasets. In summary, existing approaches to data augmentation for NLP include lexicon replacement, sentence retrieval, and round-trip machine translation (Wang & Yang, 2015; Yu et al., 2018; Kobayashi, 2018; Wu et al., 2019; Lichtarge et al., 2019; Wei & Zou, 2019; Alberti et al., 2019; Du et al., 2020; Shen et al., 2020). By contrast, we propose the use of unconditional autoregressive LMs for data augmentation. This is simple, flexible, and powerful.

We take a pretrained GPT-2 language model (Radford et al., 2019) and fine-tune it separately on each dataset of interest after removing class labels. We find that training from scratch on these datasets

is hopeless, but the larger the pretrained GPT-2 variant, the better the validation perplexity scores are. For tasks modeling a relationship between multiple sentences, we concatenate a separator token "[SEP]" between consecutive sentences. Once a fine-tuned GPT-2 model is obtained, we generate task-specific synthetic data up to $40\times$ larger than the original training sets. For some samples of generated text for GLUE see Table C.1 to C.6. We believe using bigger LMs and larger synthetic datasets will improve our results, but we are constrained by compute resources.

**Tabular data.** Features from tabular tasks are often in a well-structured format, *i.e.,* each data point comprises a fixed number of attributes as in Table D.2. This property impedes the acquisition of task-specific unlabeled data, *i.e.,* most augmentation techniques such as round-trip translation and retrieval are hardly applicable. To enable generative modeling on tabular data, we convert each data point (*i.e.,* row from the table) into a sentence by concatenating all of its attributes. This reformatting enables the use of GPT-2 fine-tuning similar to text, and surprisingly GPT-2 samples are very effective.

## 4.2 An Empirical Risk Minimization Perspective

In supervised learning, one seeks to learn a mapping $f$ that given an input $\boldsymbol{x}$, predicts a reasonable output $y$. To define the supervised learning problem formally, one assumes that input-output pairs are drawn from a joint distribution $P$, *i.e.,* $(\boldsymbol{x}, y) \sim P(\boldsymbol{x}, y)$, and a loss function $H(y, f(\boldsymbol{x}))$ is used to assess the quality of a mapping $f$. This loss is used to define a notion of *expected risk*:

$$R(f) = \mathbb{E}_{P(\boldsymbol{x},y)} H(y, f(\boldsymbol{x})) . \tag{4}$$

In almost all practical applications $P(\boldsymbol{x}, y)$ is unknown. Hence, a labeled dataset of examples $L = \{(\boldsymbol{x}_i, y_i)\}_{i=1}^N$ is used to approximate $R(f)$ as

$$\widehat{R}(f) = \frac{1}{N} \sum\nolimits_{i=1}^N H(y_i, f(\boldsymbol{x}_i)) . \tag{5}$$

This objective function is known as *empirical risk*, and learning $f$ through minimizing $\widehat{R}(f)$ is known as the *empirical risk minimization* principle (Vapnik, 1992). To compensate for the finite sample size in (5), one typically combines $\widehat{R}(f)$ with a regularizer to improve generalization.

**Beyond empirical risk minimization.** Empirical risk minimization (5) is motivated as a way to approximate $P(\boldsymbol{x}, y)$ through a set of Dirac delta functions on labeled examples: $P_\delta(\boldsymbol{x}, y) = \sum_i \delta(\boldsymbol{x} = \boldsymbol{x}_i, y = y_i)/N$. However, this approximation is far from perfect, hence one uses a heldout validation set for early stopping and hyper parameter tuning.

Vicinal risk minimization (Chapelle et al., 2001) approximates expected risk as $\mathbb{E}_{P_\nu(\boldsymbol{x},y)} H(y, f(\boldsymbol{x}))$, using a *vicinity distribution*, *e.g.,* $\nu(\tilde{\boldsymbol{x}}, \tilde{y} \mid \boldsymbol{x}, y) = \mathcal{N}(\tilde{\boldsymbol{x}} - \boldsymbol{x}, \sigma^2)\delta(\tilde{y} = y)$ to approximate $P(\boldsymbol{x}, y)$ as

$$P_\nu(\boldsymbol{x}, y) = \frac{1}{N} \sum\nolimits_{i=1}^N \nu(\tilde{\boldsymbol{x}} = \boldsymbol{x}, \tilde{y} = y \mid \boldsymbol{x}_i, y_i) . \tag{6}$$

The goal is to increase the support of each labeled data point and improve the quality and robustness of the risk function.

Recent work on mixup regularization (Zhang et al., 2018) proposes an effective way to construct another vicinity distribution by interpolating between two data points and their labels. Albeit its simplicity, these smoothing techniques tend to improve matters.

**Generative models for risk minimization.** One can factorize the joint distribution of input-output pairs as $P(\boldsymbol{x}, y) = P(\boldsymbol{x})P(y \mid \boldsymbol{x})$. Accordingly, if one is able to learn a reasonable unconditional generative model of $\boldsymbol{x}$ denoted $g(\boldsymbol{x})$, then one can draw a pair $(\boldsymbol{x}, y)$ by first drawing $\boldsymbol{x} \sim g(\boldsymbol{x})$ and then using the current instance of $f_t$ to draw $y \sim f_t(\boldsymbol{x})$. Then, one can use $f_t$ and $g$ to approximate expected risk as

$$R_t(f_{t+1}) = \mathbb{E}_{\boldsymbol{x} \sim g(\boldsymbol{x})} \mathbb{E}_{y \sim f_t(\boldsymbol{x})} H(y, f_{t+1}(\boldsymbol{x})) . \tag{7}$$

The quality of this approximation highly depends on the quality of $f_t$ and $g$. If $f_t$ is far from an optimal classifier $f^*$ or $g(\boldsymbol{x})$ is far from $P(\boldsymbol{x})$, (7) yields a poor approximation.

The expected risk in (7) smoothens the risk landscape in complex ways beyond simple Gaussian smoothing and interpolation. This smoothing is applicable to any continuous, discrete, or structured domain as long as expressive generative models of $P(\boldsymbol{x})$ are available. That said, for almost all

reasonable loss functions $H$ (*e.g.,* softmax cross entropy and squared error), (7) is minimized when $f_{t+1} = f_t$, which is not ideal, especially when $f_t$ is far from $f^*$. On the other hand, empirical risk (5) anchors the problem in real labeled examples that are provided as ground truth.

GAL aims to combine the benefits of (5) and (7) via:

$$R_t(f_{t+1}) = \frac{\lambda}{N} \sum_{i=1}^{N} H(y_i, f_{t+1}(\boldsymbol{x}_i)) + (1 - \lambda)\mathbb{E}_{\boldsymbol{x} \sim g(\boldsymbol{x})}\mathbb{E}_{y \sim f_t(\boldsymbol{x})} H(y, f_{t+1}(\boldsymbol{x})) \qquad (8)$$

In this formulation, if $f_t$ represents the minimizer of empirical risk (5), then $f_{t+1} = f_t$ is the minimizer of (8) too. However, one does not seek the global minimizer of empirical risk, but rather the best performance on heldout data. If $f_t$ is obtained by stochastic gradient descent on any risk function, but early stopped according to empirical risk on a heldout set, then using such $f_t$ in (8) to define $R_t(f_{t+1})$ promotes the selection of a mapping $f_{t+1}$ that minimizes empirical risk while staying close to the best performing mapping so far (*i.e.,* $f_t$). This formulation motivates self-training and GAL as regularizers in the functional space and explains why they can conceivably work.

**How about class-conditional generative models?** One can also factorize the joint distribution $P(\boldsymbol{x}, y)$ as $P(y)P(\boldsymbol{x} \mid y)$ and accordingly utilize a class-conditional generative model $g(\boldsymbol{x} \mid y)$ to derive the following expected risk formulation:

$$R(f) = \mathbb{E}_{y \sim P(y)}\mathbb{E}_{\boldsymbol{x} \sim g(\boldsymbol{x}|y)} H(y, f_{t+1}(\boldsymbol{x})) . \qquad (9)$$

In this setting pseudo labeling is not needed as synthetic data is already labeled. One can show that the optimal classifier $f_g^*$ that minimizes (9) for cross entropy loss is given by,

$$f_g^*(y \mid \boldsymbol{x}) = g(\boldsymbol{x}|y)P(y) \Big/ \sum_{y'} g(\boldsymbol{x}|y')P(y') , \qquad (10)$$

that is turning the class-conditional generative model into a classifier by using the Bayes rule yields the optimal solution.

Provided that the accuracy of generative classifiers on natural image and text classification is far behind their discriminate counterparts (*e.g.,* Ravuri & Vinyals, 2019), we think substituting (9) into (8) is not a good idea. Essentially, by substituting (9) into the classification objective, one is regularizing $f$ to remain close to $f_g^*$, which is not an effective strategy if $f_g^*$ is not competitive. This argument corroborates the evidence from our ablation studies and recent work showing that using class-conditional generative models to augment supervised learning does not provide big gains (Ravuri & Vinyals, 2019). That said, one can still use class-conditional generative models to synthesize high-fidelity samples. As long as these samples are treated as unlabeled examples and annotated using a classifier, *e.g.,* $f_t$, we believe this is a reasonable approach falling under GAL. Our argument above only applies to the scenario that class-conditional generative models are used to synthesize labeled examples.

## 5 EXPERIMENTS

We asses the effectiveness of GAL on KD, self-training and few-shot learning for NLP. We also present self-training results on tabular tasks. Appendix B shows the applicability of GAL to two image classification tasks as a proof of concept, but more advanced techniques such as Mixup (Zhang et al., 2018) are needed to bridge the gap with the state-of-the-art.

### 5.1 STATE-OF-THE-ART RESULTS ON KNOWLEDGE DISTILLATION ON GLUE

We use the GLUE benchmark (Wang et al., 2019b) for our KD experiments; see Appendix D for benchmark details and Appendix E for the details of synthetic text generation. Our synthetic unlabeled dataset $U$ includes $40\times$ as many examples as the original dataset for each task in the GLUE benchmark.

The goal of knowledge distillation (KD) (Buciluǎ et al., 2006; Hinton et al., 2015) is to distill the knowledge of a powerful teacher model into a compact student model with as little loss in performance as possible. This can help with model compression (Jiao et al., 2019; Sun et al., 2019a) and multi-task learning (Liu et al., 2019a; Clark et al., 2019). It is known that KD on fresh data, unseen during training, performs better (Buciluǎ et al., 2006; Chen et al., 2020c) than KD on original training data. Accordingly, we investigate the effectiveness of knowledge distillation using generated unlabeled data through GAL.

**Table 1:** GLUE test results for single 6-layer transformer models. GAL establishes a new state of the art on KD for NLP. Baselines: BERT-Theseus (Xu et al., 2020), BERT-PKD (Sun et al., 2019a), tinyBERT (Jiao et al., 2019) tinyRoBERTa (Rashid et al., 2021), DistilRoBERTa (Sanh et al., 2019), and DistilRoBERTa + KD (standard KD) and DistilRoBERTa + RT (round-trip translation to generate unlabeled text). Accuracy scores on MNLI-matched/MNLI-mismatched are reported for MNLI, Matthew's correlation is reported for CoLA, F1/Accuracy scores are reported for QQP and MRPC, Pearson/Spearman correlations are reported for STS-B, and Accuracy is reported for QNLI and RTE.

| Model | MNLI | CoLA | SST-2 | MRPC | STS-B | QQP | QNLI | RTE | Avg |
|---|---|---|---|---|---|---|---|---|---|
| *Previous work:* | | | | | | | | | |
| BERT-Theseus | 82.4/82.1 | 47.8 | 92.2 | 87.6/83.2 | 85.6/84.1 | 71.6/89.3 | 89.6 | 66.2 | 78.6 |
| BERT-PKD | 81.5/81.0 | - | 92.0 | 85.0/79.9 | - | 70.7/88.9 | 89.0 | 65.5 | - |
| tinyBERT | 84.6/83.2 | 51.1 | 93.1 | 87.3/82.6 | 85.0/83.7 | 71.6/89.1 | 90.4 | 70.0 | 79.8 |
| tinyRoBERTa | 86.2/85.6 | 58.6 | 95.1 | 91.2/88.1 | 88.5/88.4 | 73.0/89.7 | 92.4 | 76.6 | 83.5 |
| *Our results:* | | | | | | | | | |
| DistilRoBERTa | 83.8/83.4 | 55.9 | 93.2 | 87.4/83.1 | 87.5/87.5 | 71.7/89.1 | 90.6 | 73.3 | 81.2 |
| DistilRoBERTa + KD | 84.7/84.5 | 54.9 | 94.1 | 88.0/84.4 | 87.4/86.6 | 72.1/89.2 | 91.6 | 73.8 | 81.6 |
| DistilRoBERTa + RT | 86.1/86.1 | 53.0 | 94.6 | 91.0/87.8 | 89.2/88.8 | 73.1/89.9 | 92.4 | 76.9 | 82.7 |
| DistilRoBERTa + GAL | **87.4/86.5** | **60.0** | **95.3** | **91.9/89.2** | **90.0/89.6** | **73.3/90.0** | **92.7** | **81.8** | **84.8** |

**Table 2:** RoBERTa base and GAL self-training results on GLUE dev sets, averaged across 5 independent runs.

| Model | MNLI | CoLA | SST-2 | MRPC | STS-B | QQP | QNLI | RTE | Avg |
|---|---|---|---|---|---|---|---|---|---|
| RoBERTa base | 87.7 | 63.6 | 94.8 | 90.1 | 90.8 | 91.5 | 92.6 | 78.8 | 86.2 |
| + GAL (iter 1) | 87.9 | 65.1 | 95.3 | 91.7 | 91.4 | 91.8 | 93.1 | 81.4 | 87.2 |
| + GAL (iter 2) | 88.0 | 65.2 | 95.3 | 92.2 | 91.5 | 91.7 | 93.2 | 82.4 | **87.4** |
| + GAL (iter 3) | 87.9 | 65.5 | 95.3 | 92.2 | 91.7 | 91.7 | 93.2 | 82.0 | **87.4** |
| RoBERTa base + self-distillation | 88.1 | 63.7 | 95.2 | 90.3 | 90.4 | 91.5 | 93.1 | 79.7 | 86.5 |

We use the HuggingFace implementation (Wolf et al., 2020) for KD experiments and adopt a standard experimental setup consistent with previous work (Sun et al., 2019a; Xu et al., 2020). Following Rashid et al. (2021), fine-tuned RoBERTa-large (24-layer transformer) represents the teacher and a DistilRoBERTa (6-layer transformer) (Sanh et al., 2019) is used as the student. We train the student model on $U$ and $L$, where $U$ is annotated by an ensemble of 10 models, achieving an average score of 87.9. We then mix $U$ and $L$ with a ratio of 1:4, which is equivalent to $\lambda = 0.2$. This ratio works best on the dev set.

Table 1 shows the results of individual 6-layer transformers on the GLUE test set. All of the baselines use an identical student architecture. GAL achieves the best entry on the GLUE leaderboard, marking a new state-of-the-art for KD on NLP. It outperforms strong KD baselines such as DistilRoBERTa (Sanh et al., 2019), BERT-PKD (Sun et al., 2019a), BERT-Theseus (Xu et al., 2020), tinyBERT (Jiao et al., 2019) and tinyRoBERTa (Rashid et al., 2021). It also outperforms our own DistilRoBERTa+KD baseline, which learns from soft labels produced by an identical RoBERTa-large ensemble on the original labeled dataset. While the use of soft labels outperform the vanilla fine-tuned DistilRoBERTa model, it significantly underperforms our KD+GAL baseline. We also compare with round-trip translation (RT), a strong data-augmentation baseline (*e.g.,* Yu et al., 2018; Shleifer, 2019). We mirror the experimental setup of GAL and generate 40× unlabeled data using German as the bridge language (English →German→English). The translations are generated via the best model in WMT19 (Ng et al., 2019). Although DistilRoBERTa+RT is better than vanilla DistilRoBERTa and KD variants, it still significantly underperforms our approach.

## 5.2 NLP Self-Training Experiments on GLUE

We fine-tune pretrained RoBERTa models provided by fairseq (Ott et al., 2019) on each GLUE task. Fine-tuned RoBERTa serves as the first teacher model for self-training. Each student model is initialized with the original pretrained RoBERTa and fine-tuned with exactly the same hyperparameters as suggested by fairseq (Ott et al., 2019). We combine the labeled dataset $L$ and the synthetic dataset $U$ with a ratio of 1:1, by oversampling labeled data. This corresponds to $\lambda = 0.5$ in Eq. (8).

Table 2 shows that GAL provides an average improvement of +1.3% over RoBERTa-base. We see consistent improvements with more GAL iterations, but performance saturates after three iterations.

We further compare our approach with a self-distillation (Furlanello et al., 2018) baseline, in which the teacher and student models use the same architecture and transfer knowledge via the original labeled training set. Although self-distillation provides a slight improvement, the gains from GAL are more significant.

We delve deeper and combine GAL self-training with RoBERTa-large and report test results for both single model and ensemble model in Table 3. We observe consistent gains coming from GAL on RoBERTa-large. Our results underperform the latest and biggest LMs from the GLUE leaderboard, but we are optimistic that GAL can be effectively combined with enormous LMs to provide additional gains.

**Table 3:** RoBERTa-large with GAL self-training and SoTA methods evaluated on GLUE test sets. The benefit of GAL on single models is larger than ensembles. It appears that self-training reduce the variance of models. Baselines including much larger models: RoBERTa-large (Liu et al., 2019b), ELECTRA (Clark et al., 2020), T5 (Raffel et al., 2020), ERNIE (Sun et al., 2019b), and DeBERTa (He et al., 2020)

| Model | MNLI | CoLA | SST-2 | MRPC | STS-B | QQP | QNLI | RTE | Avg |
|---|---|---|---|---|---|---|---|---|---|
| *Individual Models (our implementation):* | | | | | | | | | |
| RoBERTa-large | 90.1/89.7 | 63.8 | 96.1 | 91.2/88.3 | 90.9/90.7 | 72.5/89.6 | 94.5 | 85.9 | 86.5 |
| RoBERTa-large + GAL | 90.2/89.8 | 66.2 | 96.4 | 92.0/89.2 | 90.7/90.5 | 73.6/89.9 | 95.0 | 86.3 | 87.1 |
| *Ensemble Models (our implementation):* | | | | | | | | | |
| RoBERTa-large | 91.2/90.5 | 66.8 | 96.9 | 92.8/90.3 | 91.9/91.6 | 74.5/90.4 | 95.5 | 87.7 | 87.9 |
| RoBERTa-large + GAL | 91.0/90.7 | 67.9 | 97.1 | 93.1/90.8 | 91.6/91.4 | 74.5/90.4 | 95.8 | 88.2 | 88.2 |
| *State-of-the-art:* | | | | | | | | | |
| RoBERTa-large | 90.8/90.2 | 67.8 | 96.7 | 92.3/89.8 | 92.2/91.9 | 74.3/90.3 | 95.4 | 88.2 | 88.0 |
| ELECTRA | 91.3/90.8 | 71.7 | 97.1 | 93.1/90.7 | 92.9/92.5 | 75.6/90.8 | 95.8 | 89.8 | 89.2 |
| T5 | 92.2/91.9 | 71.6 | 97.5 | 92.8/90.4 | 93.1/92.8 | 75.1/90.6 | 96.9 | 92.8 | 89.8 |
| ERNIE | 91.9/91.4 | 74.4 | 97.8 | 93.9/91.8 | 93.0/92.6 | 75.2/90.9 | 97.3 | 92.0 | 90.2 |
| DeBERTa | 91.9/91.6 | 71.5 | 97.5 | 94.0/92.0 | 92.9/92.6 | 76.2/90.8 | 99.2 | 93.2 | 90.3 |

## 5.3 Self-Training on Tabular Tasks

We assess the effectiveness of GAL self-training on four tabular tasks, namely connect-4 (Burton & Kelly, 2006), Drug Review (Gräßer et al., 2018), Drybean (Koklu & Ozkan, 2020) and Spambase (Dua & Graff, 2017). The details of these tasks can be found in Appendix D. We follow the same protocol as GLUE tasks and generate $40\times$ unlabeled data from a fine-tuned GPT-2-large. Table 4 shows that GAL achieves sizable gains on these tasks even though neither RoBERTa nor GPT-2 are optimized for tabular tasks. XGBoost (Chen & Guestrin, 2016), a strong supervised baseline for tabular tasks underperforms RoBERTa+GAL on connect-4 and Drug Review. It is worth noting that since inputs of Drug Review contain free form text, we convert them into numeric values through the max-pooled representation of the last hidden states of RoBERTa base. XGBoost outperforms RoBERTa on Drybean and Spambase, but it is important to note that these two datasets include floating point attributes, which we simply tokenize using BPE (Sennrich et al., 2016) into subwords. Nevertheless, GAL is capable of bridging the gap between transformers and XGBoost. We believe GAL can be successfully combined with XGBoost, but we leave this to future work, since the XG-Boost library does not easily accommodate soft labels.

**Table 4:** RoBERTa-base and GAL results on four tabular datasets from the UCI repository. Accuracy is reported for these datasets.

| Model | connect-4 | Drug Review | Drybean | Spambase |
|---|---|---|---|---|
| XGBoost | 86.0 | 80.1 | 92.1 | 96.7 |
| RoBERTa base | 85.0 | 84.6 | 85.0 | 87.7 |
| + GAL (iter 1) | 87.0 | 85.7 | 85.8 | 89.0 |
| + GAL (iter 2) | 87.5 | 85.8 | 86.0 | 88.8 |
| + GAL (iter 3) | 87.3 | 85.6 | 85.9 | 89.3 |

**Table 5:** Few-shot learning results for GPT-J (6B) (Wang & Komatsuzaki, 2021) on four NLP datasets. Accuracy is reported for these datasets.

| Model | SST-2 | PIQA | COPA | BoolQ | Avg |
|---|---|---|---|---|---|
| 4-shot | 89.8 | 76.0 | 79.0 | 64.3 | 77.3 |
| 8-shot | 91.3 | 76.2 | 79.0 | 66.2 | 78.2 |
| 16-shot | 92.7 | 77.0 | 81.0 | 66.8 | 79.4 |
| 4-shot + synthetic 12-shot (GAL) | 91.5 | 76.7 | 80.0 | 65.9 | 78.5 |

## 5.4 PROMPT-BASED FEW-SHOT EXPERIMENTS

GPT3 (Brown et al., 2020) has introduced an optimization-free paradigm for few-shot learning for NLP. Without updating the parameters, large LMs can correctly predict the labels of the inputs by conditioning on a prompt, which consists of an instruction, a few labeled instances and a new unlabeled input. We apply GAL to prompt-based few-shot learning. Specifically, we present $k$ labeled examples as a prompt to GPT-J (Wang & Komatsuzaki, 2021), an open-sourced re-implementation of GPT-3-6B, and generate $m$ synthetic examples, followed by the corresponding labels. Note that to mitigate noisy outputs, the generation of each synthetic example only conditions on the original $k$ labeled examples. Finally, we concatenate the original $k$ examples and $m$ synthetic examples, and conduct a $(k + m)$-shot learning experiment with GPT-J.

Brown et al. (2020) studied a total of 51 few-shot learning tasks. Studying all of these tasks is prohibitively expensive. Thus, we filter tasks by following these two steps. First, since generating $m$ synthetic examples for each test instance is computationally expensive, we exclude tasks that have more than 5k test examples. Second, we filter tasks on which GPT-3-6B achieves a score lower than 65% (please refer to Table H.1 in Brown et al. (2020) for more details). After applying the filtering steps, we use four datasets: SST-2 (Wang et al., 2019b), PIQA (Bisk et al., 2020), COPA and BoolQ (Wang et al., 2019a) as the testbed. We notice that in order to generate valid synthetic data, GPT-J requires to see at least 4 labeled examples. In addition, at most 16 examples of BoolQ can be fed into GPT-J without truncation. Thus, we set $k$ and $m$ to 4 and 12 respectively. As seen in Table 5, GAL leads to an average improvement of 1.2% over 4-shot learning, and reduces the gap between 4-shot and 16-shot learning. We noticed that the quality of some generated examples is low. We believe the performance of few-shot learning can be further improved with high-quality instances. One solution is to generate many synthetic examples, and select a high-quality subset. Since each test instance conditions on distinct labeled instances, one has to generate different synthetic instances for each test example from GPT-J, which causes expensive computation. Due to such computational constraints, we leave the investigation of data selection strategies to the future work.

## 5.5 ABLATING COMPONENTS OF GAL ON GLUE

We conduct an in-depth study of different components of GAL on GLUE datasets. Unless stated otherwise, we use a RoBERTa-base model with a combination of the original training data and $40\times$ synthetic data for each experiment.

**Table 6:** GAL with various GPT-2 model sizes on GLUE dev sets. NA indicates a vanilla RoBERTa base model.

| GPT-2 | SST-2 | RTE | MRPC | CoLA |
|---|---|---|---|---|
| NA | 94.8 | 78.8 | 90.1 | 63.6 |
| small | 95.5 | 81.3 | 90.9 | 63.9 |
| medium | 95.3 | 81.3 | 91.3 | 63.7 |
| large | 95.3 | 81.4 | 91.7 | 65.1 |

**GPT-2 model size.** Radford et al. (2019) present a few variants of the GPT-2 model including *GPT-2*, *GPT-2-medium*, *GPT-2-large*, and *GPT-2-XL*. Larger GPT-2 models yield better perplexity scores and higher generation quality. We utilize these models except GPT-2-XL within the GAL framework to study the impact of the generative model's quality on downstream task's performance. Table 6 shows that regardless of the GPT-2 model sizes, GAL consistently surpasses the vanilla RoBERTa base. Moreover, SST-2 and RTE datasets are not sensitive to the capacity of the GPT-2 model, but higher quality synthetic text improves the results on MRPC and CoLA datasets. We leave investigation of GPT-2-XL and even larger LMs such as GPT-3 (Brown et al., 2020) to future work.

**Class-conditional synthetic data generation.** Previous work (Kumar et al., 2020b; Ravuri & Vinyals, 2019) suggests that it is challenging to utilize synthetic data from class-conditional gener-

ative models to boost the accuracy of text and image classifiers. Our theory in Section 4.2 points to the potential drawback of class-conditional synthetic data. We empirically study this phenomenon, by fine-tuning GPT-2 in a class-conditional manner. Table 7 shows that not only class-conditional LMs underperform unconditional LMs in our GAL framework, but also they are much worse than the baseline.

**Table 7:** Synthetic data from class-conditional LMs underperforms GAL and RoBERTa on GLUE dev sets.

| Source of synthetic data | SST-2 | RTE | MRPC | CoLA |
|---|---|---|---|---|
| No synthetic data (baseline) | 94.8 | 78.8 | 90.1 | 63.6 |
| Class-conditional LM | 92.9 | 74.4 | 86.0 | 58.4 |
| Unconditional LM (GAL) | 95.3 | 81.4 | 91.7 | 65.1 |

## 6 CONCLUSION

We present Generate, Annotate, and Learn (GAL): a framework for self-training and knowledge distillation with generated unlabeled data. We motivate GAL from an expected risk minimization perspective and demonstrate both theoretically and empirically that the use of unconditional generative models for synthetic data generation is more effective than class-conditional generative models, previously used in the literature. GAL leverages advances in large pretrained language models to help supervised learning and can have implications for learning from limited labeled data. GAL works surprisingly well on NLP and tabular tasks, and helps improve knowledge distillation and prompt-based few-shot learning. We hope that GAL will stimulate new research on the evaluation and development of large language models.

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

## A  PSEUDO-CODE OF ALGORITHMS

---

**Algorithm 1** SelfTraining($L, U, f_0, T$)

---

**Input:** Labeled dataset $L = \{(\boldsymbol{x}_i, y_i)\}_{i=1}^N$
  Unlabeled dataset $U = \{\boldsymbol{x}_j\}_{j=1}^M$
  Initial parameters of a classifier $f_0$
**Output:** A better classifier $f_{T+1}$ after $T$ self-training steps
  1: train a base model $f_1$ by fine-tuning $f_0$ on $L$
  2: **for** $t = 1$ to $T$ **do:**
  3:    apply $f_t$ to unlabeled instances of $U$
  4:    select a subset $S_t \subseteq \{(x, f_t(x)) \mid x \in U\}$
  5:    train a new model $f_{t+1}$ by either fine-tuning $f_0$ on $L \cup S_t$
       or gradient descend on a minibatch from $L \cup S_t$
  6: **return** $f_{T+1}$

---

---

**Algorithm 2** GAL($L, g_0, f_0, k, T$)

---

**Input:** Labeled dataset $L = \{(\boldsymbol{x}_i, y_i)\}_{i=1}^N$
  Initial parameters of a generative model $g_0$
  Initial parameters of a classifier $f_0$
**Output:** A better classifier $f_{T+1}$ after $T$ GAL steps
  1: train a generative model $g$ by fine-tuning $g_0$ on $L_x$ where $L_x = \{\boldsymbol{x} \mid (\boldsymbol{x}, y) \in L\}$
  2: generate $U = \{\widetilde{\boldsymbol{x}}_j\}_{j=1}^{kN}$ by drawing $kN$ random samples *i.i.d.* from $g(\boldsymbol{x})$, *i.e.,* $\widetilde{\boldsymbol{x}}_j \sim g(\boldsymbol{x})$ for $j = 1$ to $kN$.
  3: **return** SelfTraining($L, U, f_0, T$)

---

## B  GAL ON IMAGE CLASSIFICATION TASKS

As a proof of concept, in addition to NLP and tabular tasks, we assess the effectiveness of GAL on CIFAR-10 (Krizhevsky & Hinton, 2009) and Fashion MNIST (Xiao et al., 2017) as well. We adopt the NCSN model of Song & Ermon (2019) as the task-specific generative model. We use the CIFAR-10 model provided by the authors and train a model on Fashion MNIST using the same configuration as CIFAR-10. We select the model checkpoint resulting in the best FID score vs. training set (Heusel et al., 2017) based on 1000 samples. We then use the NCSN models to generate up to $10\times$ synthetic unlabeled data, *i.e.,* 500K for CIFAR-10 and 600K for Fashion MNIST. See Appendix C for representative samples.

We adopt FixMatch (Sohn et al., 2020) to conduct semi-supervised learning on vision tasks, since Fix-Match has shown promising results on CIFAR-10. Specifically, we train a classifier on mini-batches of intertwined labeled and unlabeled data (synthetic). In each iteration, we obtain pseudo-labels for the unlabeled data, but filter unlabeled examples based on classifier's confidence, *i.e.,* examples are kept on which the largest class probability exceeds $\tau$. Weak augmentation is used to define pseudo labels, but strong augmentations are used to obtain student model's predictions. We randomly sample from the strong augmentations list defined in RandAugment (Cubuk et al., 2020). We only apply strong augmentations to the synthetic samples and not the original labeled data to ensure a fair comparison with the baseline.

**Table B.1:** Classification error rates on CIFAR-10 test set with varying amounts of synthetic data for three different model architectures. Reported results are the average of 3 independent runs.

| Model      | VGG19 | ResNet110 | WRN28-10 |
|------------|-------|-----------|----------|
| # params   | 1.74M | 20.11M    | 36.48M   |
| Baseline   | 6.62  | 5.85      | 3.87     |
| GAL 1×     | 5.97  | 5.13      | 3.75     |
| GAL 5×     | 5.80  | 5.11      | 3.25     |
| GAL 10×    | **5.65** | **5.10** | **3.23** |

We conduct experiments on three different convolutional neural network architectures: VGG19 (Simonyan & Zisserman, 2014), WideResnet28-10 (Zagoruyko & Komodakis, 2016), and ResNet110 (He et al., 2016). For the full list of hyperparameters and other implementation details, please refer to Appendix H. Each classifier is trained for 200 epochs and 3 synthetic datasets of size ($1\times$, $5\times$, $10\times$) of the training dataset are used.

Table B.1 shows that GAL achieves an average error reduction of $0.78\%$ over the baseline on CIFAR-10 across the 3 architectures tested. Further, it appears that the larger the synthetic dataset size, the better the performance of GAL is. We note that the reported results are the average of 3 independent runs. Similarly on Fashion MNIST, we witness consistent gains across all architectures. Fashion MNIST results are included in Appendix I. Our image classification experiments confirm that even when the generative model of GAL is not pretrained on open domain data and solely trained on the dataset at hand, GAL can offer significant improvements.

Table B.2 presents GAL results on Fashion MNIST dataset. Similar to CIFAR-10, we observe a performance improvement across the three architectures.

**Table B.2:** Classification error rates on Fashion MNIST test set with varying amounts of synthetic data for three different model architectures. Results reported are the average over 3 independent runs.

| **Model** | VGG19 | WRN28-2 | ResNet110 |
|---|---|---|---|
| **# params** | 1.74M | 1.98M | 20.11M |
| Baseline | 5.41 | 4.92 | 5.21 |
| GAL $1\times$ | 5.06 | **4.63** | **4.74** |
| GAL $5\times$ | 5.14 | 4.85 | 4.85 |
| GAL $10\times$ | **4.90** | 4.74 | 4.75 |

## C    GENERATED UNLABELED EXAMPLES ANNOTATED WITH PSEUDO LABELS

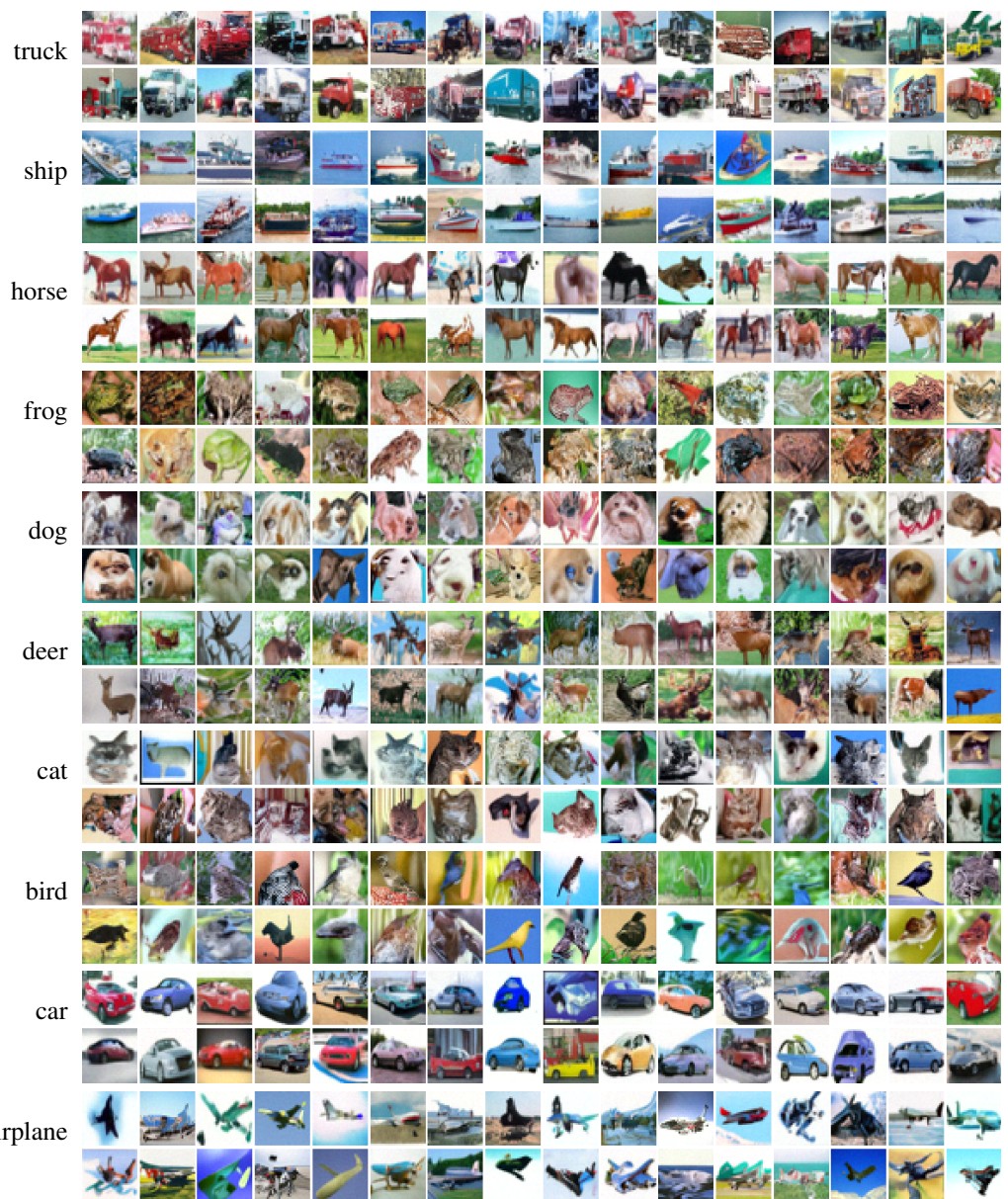

**Figure C.1:** CIFAR-10 synthetic samples generated by NCSN (Song & Ermon, 2019) and corresponding pseudo-labels. Images are filtered based on a confidence threshold of $\tau = 0.95$ and categorized based on pseudo-labels. For each category, 16 random samples are shown.

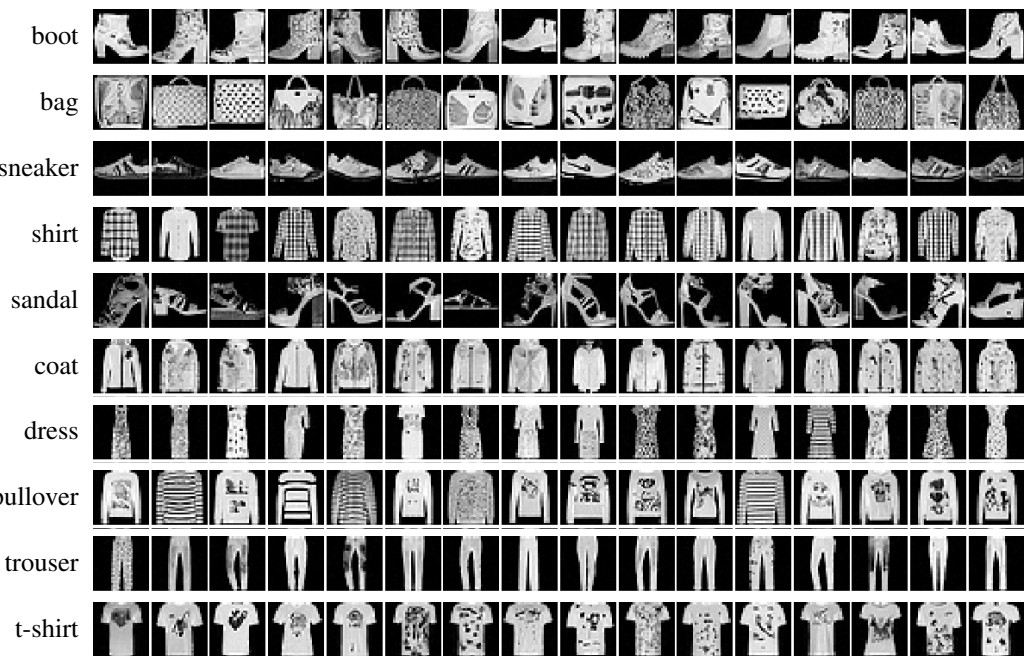

**Figure C.2:** Fashion MNIST synthetic samples generated by NCSN (Song & Ermon, 2019) and pseudo-labels. Images are filtered based on a confidence threshold of $\tau = 0.95$ and categorized based on pseudo-labels. For each category, 16 random samples are shown.

**Table C.1: QNLI**: Two labeled examples, along with 3 nearest neighbors (based on RoBERTa representations) from our synthetic dataset. We include **labels** for original examples and **pseudo-labels** for synthetic examples in parenthesis.

| |
|---|
| When did the third Digimon series begin? [SEP] Unlike the two seasons before it and most of the seasons that followed, Digimon Tamers takes a darker and more realistic approach to its story featuring Digimon who do not reincarnate after their deaths and more complex character development in the original Japanese. (**not entailment**) |
| KNN:
1: What is the name of the third season? [SEP] In addition to the first two seasons, the third season is the season that introduced new characters such as Captain Malice, a supervillain who became the antagonist in season two; and the villains known as the Heartbreakers, who introduced a group of crime fighters. (**not entailment**)
2: When did the "Walking Dead" series end? [SEP] In 2013, AMC announced that it would develop a "superhero series", which would follow the storylines and characters from the "Walking Dead" series in order to bring the popular AMC original series to a new and younger audience. (**not entailment**)
3: What is the main objective of the first season of the X-Files? [SEP] The first season was notable in that the characters were introduced and developed within the space of a single season, as was the format of the show itself. (**not entailment**) |
| What did Arsenal consider the yellow and blue colors to be after losing a FA Cup final wearing red and white? [SEP] Arsenal then competed in three consecutive FA Cup finals between 1978 and 1980 wearing their "lucky" yellow and blue strip, which remained the club's away strip until the release of a green and navy away kit in 1982–83. (**entailment**) |
| KNN:
1: Who was the most important player for Arsenal Football Club in the 1950s? [SEP] Wenger continued to use Arsenal's famous red shirts and red kits throughout the 1950s and 1960s, and the red strip became the club's most recognised and recognizable symbol. (**not entailment**)
2: When were the first two teams to play for the trophy in the Premier League? [SEP] The trophy was awarded to Manchester United in 1990-91 and was named after Sir Bobby Charlton, the club's manager until 1990, and later Sir Stanley Matthews, the club's most successful manager. (**not entailment**)
3: What were the last four players to wear the yellow in the final? [SEP] With Arsenal having won all four major trophies in the period, they became the only club to have won five in a row. (**not entailment**) |

**Table C.2: QQP**: Two labeled examples, along with 3 nearest neighbors (based on RoBERTa representations) from our synthetic dataset. We include **labels** for original examples and **pseudo-labels** for synthetic examples in parenthesis.

| |
|---|
| How is the life of a math student? Could you describe your own experiences? [SEP] Which level of prepration is enough for the exam jlpt5? (**not duplicated**) |
| KNN:
1: What are the best courses for a mechanical engineering student? [SEP] What is the best course to do after completing a B.Tech in mechanical engineering? (**not duplicated**)
2: How much marks are needed to get through the GATE with electronics? [SEP] What is the average score of the Gate EE exam? What are the cut-offs? (**not duplicated**)
3: What is the best time table for students to prepare for IAS? [SEP] How can one study for IAS in a best time? (**not duplicated**) |
| How does an IQ test work and what is determined from an IQ test? [SEP] How does IQ test works? (**duplicated**) |
| KNN:
1: What is the average IQ of the U.S. population? [SEP] How does an IQ test work? (**not duplicated**)
2: Is the Iq test an effective way to measure intelligence? [SEP] How do IQ tests work? (**duplicated**)
3: How is an IQ test on a scale from 1 to 100 scored? [SEP] How do you get your IQ tested? (**not duplicated**) |

**Table C.3: RTE**: Two labeled examples, along with 3 nearest neighbors (based on RoBERTa representations) from our synthetic dataset. We include **labels** for original examples and **pseudo-labels** for synthetic examples in parenthesis.

| |
|---|
| Like the United States, U.N. officials are also dismayed that Aristide killed a conference called by Prime Minister Robert Malval in Port-au-Prince in hopes of bringing all the feuding parties together. [SEP] Aristide had Prime Minister Robert Malval murdered in Port-au-Prince. (**not entailment**) |
| KNN:
1: The government has been criticized for failing to prevent the mass protests that led to the ouster of President Nicolas Sarkozy earlier this month, which led to his second election defeat since assuming office two years ago. [SEP] Prime Minister Jean-Marc Ayrault is a former president of France. (**not entailment**)
2: The French president, Jacques Chirac, has been urged by both the Vatican and the U.N. Security Council to step up efforts to prevent the return of former dictator Nicolas Sarkozy. [SEP] Nicolas Sarkozy left France. (**not entailment**)
3: The French newspaper Le Monde says the French President Nicolas Sarkozy was advised by U.S. President George W. Bush about a possible trip to Iraq on Thursday. [SEP] Nicolas Sarkozy is a member of the United States. (**not entailment**) |
| Only a week after it had no comment on upping the storage capacity of its Hotmail e-mail service, Microsoft early Thursday announced it was boosting the allowance to 250MB to follow similar moves by rivals such as Google, Yahoo, and Lycos. [SEP] Microsoft's Hotmail has raised its storage capacity to 250MB. (**entailment**) |
| KNN:
1: The company, known as Microsoft Office, said it plans to sell all of the copies of its popular Office suite at a loss in the wake of the launch of Microsoft Windows 7, saying it will also make $25 million in advertising costs, a move likely to hurt its long-standing position among consumers and business leaders. [SEP] Microsoft Office is a popular office suite. (**entailment**)
2: The company's shares shot up more than 35% after the company said it has sold all of its remaining inventory of the new Kindle e-readers at $70 each. The shares rose to $65.20 on Wednesday, their highest since March 6, 2011. "The Kindle is our best selling product," said Jeff Bezos, founder and CEO of Amazon.com. [SEP] Amazon.com is based in Seattle. (**not entailment**)
3: In response to concerns expressed by some investors, Microsoft last week said it would reduce the amount of shares that will be available to the public by 10 percent in the first quarter, with a further reduction to 3 percent in the second quarter. The stock price has plunged from $24 to $17, and Microsoft is currently offering $17 to $19 a share to its most senior employees. Some investors had criticized Microsoft's response to concerns about the price of its stock and about the perception that the company is in trouble. [SEP] Microsoft is struggling to sell its stock. (**not entailment**) |

**Table C.4: MRPC**: Two labeled examples, along with 3 nearest neighbors (based on RoBERTa representations) from our synthetic dataset. We include **labels** for original examples and **pseudo-labels** for synthetic examples in parenthesis.

| |
|---|
| A BMI of 25 or above is considered overweight ; 30 or above is considered obese . [SEP] A BMI between 18.5 and 24.9 is considered normal , over 25 is considered overweight and 30 or greater is defined as obese . (**paraphrase**) |
| KNN: 
 1: The report said that the average woman in her twenties who takes oral contraceptives daily can expect a loss of around 40 per cent of her bone density between the ages of 20 and 45 . [SEP] The study said the average woman in her twenties who used the pill every day , or every day for up to five years , can expect a loss of about 40 per cent of her bone density between the ages of 20 and 45 . (**paraphrase**) 
 2: The report found that 17 percent of U.S. adults between ages 18 and 64 have a body mass index at or above the ” normal ” 20 . [SEP] For people of that age , 17.1 percent of adults have a body mass index at or above the ” normal ” 20 , while 12.6 percent have a body mass index of 30 or above . (**not paraphrase**) 
 3: The survey shows the proportion of women between 20 and 44 who were obese was 6.3 percent , up from 5.7 percent in 2001 . [SEP] The proportion of women between 20 and 44 who were obese increased to 6.3 percent from 5.7 percent in 2001 . (**paraphrase**) |
| Shares of Genentech , a much larger company with several products on the market , rose more than 2 percent . [SEP] Shares of Xoma fell 16 percent in early trade , while shares of Genentech , a much larger company with several products on the market , were up 2 percent .(**not paraphrase**) |
| KNN: 
 1: Shares in Aventura fell as much as 5 percent , while shares in Medi-Cal climbed 2.5 percent . [SEP] Shares in Aventura were up 2.5 percent , while shares in Medi-Cal rose 2.5 percent . (**paraphrase**) 
 2: Shares of Amgen rose $ 2.29 , or 2.2 percent , to $ 41.10 in after-hours trading . [SEP] Shares of Amgen , a division of Sanofi-Aventis , rose $ 1.62 , or 1.6 percent , to $ 41.06 in after-hours trading .(**paraphrase**) 
 3: Shares of General Electric Co . GE.N rose more than 6 percent on the New York Stock Exchange , while shares of PepsiCo Inc . PEP.N rose 4.7 percent . [SPE] General Electric ’s shares jumped almost 6 percent on the New York Stock Exchange , while PepsiCo ’s climbed 4.7 percent . (**paraphrase**) |

**Table C.5: MNLI**: Two labeled examples, along with 3 nearest neighbors (based on RoBERTa representations) from our synthetic dataset. We include **labels** for original examples and **pseudo-labels** for synthetic examples in parenthesis.

| |
|---|
| One of our number will carry out your instructions minutely. [SEP] A member of my team will execute your orders with immense precision. (**entailment**) |
| KNN:
1: We are at your disposal to help you with your investigation and provide a full range of pro bono services. [SEP] We are the only ones who can help you with your investigation. (**neutral**)
2: I will speak with the chief officer of the contractor, who will be informed about the results of this effort. [SEP] The contractor is being informed about the results of the effort. (**entailment**)
3: We have an office here to assist you. [SEP] An office is where we will assist you, said the manager. (**neutral**) |
| Conceptually cream skimming has two basic dimensions - product and geography. [SEP] Product and geography are what make cream skimming work. (**neutral**) |
| KNN:
1: There are two main types of analysis and they are the case study and the case report. [SEP] The case study is the most popular method used to analyze a subject. (**neutral**)
2: A third approach to capturing and using this type of experience is to engage the program management and finance systems of the organization. [SEP] There are two strategies to capturing and using experience. (**contradiction**)
3: The first is to see the basic elements of a business model in action. [SEP] Basic elements of business models are the most important for the success of any company. (**neutral**) |
| I don't mean to be glib about your concerns, but if I were you, I might be more concerned about the near-term rate implications of this $1. [SEP] I am concerned more about your issues than the near-term rate implications. (**contradiction**) |
| KNN:
1: I'm not here to tell you of my own experiences, but they are important to others who might have similar concerns. [SEP] If you were to have similar concerns, I'd like to encourage you to tell them to me. (**neutral**)
2: I don't mean to sound judgmental, but as a person, I think that's an issue you're probably pretty much on your own if you think about it. [SEP] You're probably right if you think about it. (**neutral**)
3: But I don't mean to take your word for it. [SEP] I know you are correct, but I want to make sure it's clear that I do not agree. (**contradiction**) |

**Table C.6: SST-2**: Two labeled examples, along with 3 nearest neighbors (based on RoBERTa representations) from our synthetic dataset. We include **labels** for original examples and **pseudo-labels** for synthetic examples in parenthesis.

| |
|---|
| are more deeply thought through than in most ' right-thinking ' films (**positive**) |
| KNN:
1: is far more sophisticated , insightful and thought-provoking than his previous films . (**positive**)
2: is more sophisticated than its more obvious and less-than-dazzling counterparts (**positive**)
3: is about as well-thought as the idea of a bad hair day , (**negative**) |
| contains no wit , only labored gags (**negative**) |
| KNN:
1: lacks insight , and lacks empathy (**negative**)
2: has little humor or intelligence (**negative**)
3: lacks all wit and humanity (**negative**) |

# D   DATASETS

**Table D.1:** Summary of the three sets of tasks used for evaluation of GAL. STS-B is a regression task, so #classes is not applicable.

| Dataset | task | domain | #train | #dev | #test | #classes |
|---|---|---|---|---|---|---|
| NLP - GLUE Benchmark: | | | | | | |
| SST-2 | sentiment analysis | movie reviews | 67k | 872 | 1.8k | 2 |
| QQP | paraphrase | social QA questions | 364k | 40k | 391k | 2 |
| QNLI | QA/natural language inference | Wikipedia | 105k | 5k | 5.4k | 2 |
| RTE | natural language inference | news, Wikipedia | 2.5k | 277 | 3k | 2 |
| MNLI | natural language inference | misc. | 393k | 20k | 20k | 3 |
| MRPC | paraphrase | news | 3.7k | 408 | 1.7k | 2 |
| CoLA | acceptability | misc. | 8.5k | 1043 | 1k | 2 |
| STS-B | sentence similarity | misc. | 5.8k | 15k | 1.4k | – |
| Tabular Data - UCI: | | | | | | |
| connect-4 | utility value | gaming | 54k | 6.8k | 6.8k | 3 |
| Drug Review | sentiment analysis | medical | 2.6k | 0.5k | 1k | 3 |
| Drybean | categorical classification | grain | 10.9k | 1.4k | 1.4k | 7 |
| Spambase | spam classification | e-mail | 3.7k | 0.5k | 0.5k | 2 |
| Computer Vision: | | | | | | |
| CIFAR-10 | image classification | real images | 50K | N/A | 10K | 10 |
| Fashion MNIST | image classification | clothing - grey scale | 60K | N/A | 10K | 10 |

**Table D.2:** 3 examples of input and labels for the Drybean tabular task.

| | Attributes | Label |
|---|---|---|
| 1 | 37316 , 718.059 , ... , 0.6738775377027459 , 0.9981482213213235 | SIRA |
| 2 | 50634 , 892.3919999999999 , ... , 0.48054883366111584 , 0.9942734696473365 | HOROZ |
| 3 | 33631 , 669.076 , ... , 0.8466656241160356 , 0.9981796305487345 | SEKER |

# E   GENERATING SYNTHETIC TEXT FOR GLUE

To generate domain-specific synthetic data, we fine-tune GPT-2-large on the training set of each downstream task, excluding labels. For tasks with multiple input sentences, we concatenate input sentences into a long sequences and separate sentences by special [SEP] tokens. We generate new domain-specific data by using top-k random sampling similar to Radford et al. (2019). We do not feed any prompt to the LM, but a special [BOS] token to initiate the generation chain. A generation episode is terminated when a special [EOS] token is produced. We generate diverse sentences by varying the random seed. After collecting enough synthetic data, we only retain unique sentences. For tasks with $\alpha$ input sentences, we discard generated samples that violate this constraint (approximately 10% of samples were rejected).

**Quality of synthetic dataset.** An effective generative model of text should learn the word preferences and genre associated with a given corpus, but still produce novel sentences. In order to study the characteristics of our synthetic datasets, Table E.1 reports the number of unique n-grams in the training and synthetic datasets, as well as the number of unique n-grams shared between them. The high degree of overlap on uni-grams suggests that the fine-tuned GPT-2-large is somewhat domain-specific. Meanwhile, the large number of unique n-grams in the synthetic dataset suggests that many novel word combinations are generated, which is helpful for GAL.

**Table E.1:** For each dataset we report the number of unique n-grams in (the original dataset, the synthetic dataset, shared between the two).

| | SST-2 | QNLI | RTE | MRPC | CoLA |
|---|---|---|---|---|---|
| 1-gram | (15k, 33k, 11k) | (89k, 231k, 55k) | (18k, 34k, 13k) | (15k, 27k, 10k) | (6k, 6k, 4k) |
| 3-gram | (107k, 2M, 38k) | (2M, 10M, 513k) | (120k, 750k, 30k) | (105k, 562k, 27k) | (39k, 198k, 14k) |
| 5-gram | (109k, 4M, 9k) | (2M, 25M, 146k) | (130k, 1M, 4k) | (120k, 1M, 7k) | (35k, 389k, 5k) |

## F  IMPORTANCE OF PSEUDO-LABELS

We have argued and demonstrated that using class-conditional generative models to generate *labeled* synthetic examples is less effective than GAL in section 4 and section 5. To further verify this argument, we sample 100 instances from the synthetic RTE dataset generated by a class-conditional LM. Then we annotate these examples using a human annotator and the RoBERTa classifier. Finally, we compute the Accuracy, F1, Precision and Recall scores between human labels and RoBERTa labels, and between human labels and conditioning labels (i.e., labels that the class-conditional LM conditions on.). Table F.1 shows that class-conditional LM has difficulty generating sentences retaining the semantics or pragmatics of a specified category, which also corroborates our theoretical analysis in section 4. On the other hand, RoBERTa is able to produce higher quality labels that correlate better with human annotations.

**Table F.1:** Performance of RoBERTa annotation and conditioning labels on 100 random examples from the synthetic RTE dataset generated by a class-conditional LM.

| Label type | Accuracy | F1 | Precision | Recall |
|---|---|---|---|---|
| RoBERTa | 90.0 | 91.4 | 100.0 | 84.1 |
| conditioning label | 72.0 | 71.4 | 66.0 | 77.8 |

## G  GPT-2 FOR CLASSIFICATION

We have conducted additional experiments, where we fine-tune GPT-2 as a classifier. We have considered two variants of the GPT-2 model. The first varant is the original GPT-2 model (GPT2-original) pre-trained on open-domain text. The second variant is the GPT-2 model that was fine-tuned on the inputs of each task separately (GPT-2-finetuned). This model was used to generate task-specific (synthetic) unlabeled data. Finally, we also consider self-training with GAL on top of GPT2-original. Specifically, we use the GPT-2-finetuned model to synthesize 40x in-domain unlabeled data. Then we apply self-training to GPT-2-original, where the data is a combination of the original labeled data and pseudo-labeled synthetic data. Table G.1 suggests that the gains of GAL come from the pseudo-labeled synthetic data, *i.e.,* both synthetic unlabeled data and teacher's knowledge. Without the generation of synthetic unlabeled data, the domain-specific knowledge embedded in GPT-2-finetuned model cannot be utilized. As such, GPT-2-finetuned model is inferior to the GPT2-original model.

**Table G.1:** GLUE test results of different GPT-2 models.

| Model | MNLI | CoLA | SST-2 | MRPC | STS-B | QQP | QNLI | RTE | Avg |
|---|---|---|---|---|---|---|---|---|---|
| GPT-2-original | 85.9/85.6 | 54.8 | 94.5 | 86.9/82.2 | 86.3/85.2 | 72.5/89.3 | 91.2 | 69.8 | 80.9 |
| GPT-2-finetuned | 85.8/85.5 | 40.9 | 94.5 | 87.0/81.0 | 85.6/84.3 | 71.4/88.5 | 91.5 | 69.0 | 78.8 |
| GPT-2-original+GAL | 86.2/85.8 | 55.7 | 94.7 | 87.9/83.4 | 86.9/85.9 | 72.6/89.4 | 91.9 | 70.6 | 81.5 |

## H  TRAINING DETAILS

We use the fairseq codebase (Ott et al., 2019) for implementing both NLP and tabular experiments. Training details are summarized in Table H.1 and  Table H.2. We use the HuggingFace codebase (Wolf et al., 2020) for KD experiments. All NLP models are trained for 5 epochs with a learning rate of 2e-5 and a batch size of 32. All experiments are run on a single Nvidia V100 GPU.

For the CV tasks, we first use the official implementation of NCSN  (Song & Ermon, 2019) to generate the synthetic images for CIFAR-10 and Fashion MNIST. We use the pretrained checkpoints provided by the authors for the generation of synthetic CIFAR-10 images and we train a new generative model for Fashion MNIST from scratch with the same hyperparameters of the CIFAR-10 network. After generating the synthetic images, we apply GAL using a FixMatch-like setup (Sohn et al., 2020), using the hyperparameters listed in Table H.4. We follow  Cubuk et al.

**Table H.1:** Training details for NLP tasks.

|  | MNLI | CoLA | SST-2 | MRPC | STS-B | QQP | QNLI | RTE |
|---|---|---|---|---|---|---|---|---|
| lr | 1e-5 | 1e-5 | 1e-5 | 1e-5 | 2e-5 | 1e-5 | 1e-5 | 2e-5 |
| #sent. | 32 | 16 | 32 | 16 | 16 | 32 | 32 | 16 |
| warmup steps | 7432 | 320 | 1256 | 137 | 214 | 28318 | 1986 | 122 |
| validate steps | 12386 | 535 | 2093 | 203 | 360 | 11307 | 3310 | 203 |
| #epoch | 2 | 2 | 2 | 2 | 2 | 2 | 2 | 2 |

**Table H.2:** Training details for tabular tasks.

|  | connect4 | Drug | Drybean | Spambase |
|---|---|---|---|---|
| lr | 1e-5 | 1e-5 | 1e-5 | 1e-5 |
| #sent. | 32 | 16 | 16 | 16 |
| warmup steps | 1013 | 116 | 408 | 138 |
| validate steps | 1686 | 212 | 681 | 231 |
| #epoch | 2 | 2 | 2 | 4 |

**Table H.3:** Training details for KD on GLUE benchmark.

|  | MNLI | CoLA | SST-2 | MRPC | STS-B | QQP | QNLI | RTE |
|---|---|---|---|---|---|---|---|---|
| lr | 2e-5 | 2e-5 | 2e-5 | 2e-5 | 2e-5 | 2e-5 | 2e-5 | 2e-5 |
| #sent. | 32 | 32 | 32 | 32 | 16 | 32 | 32 | 16 |
| validate steps | 24542 | 534 | 4208 | 208 | 358 | 22740 | 6546 | 154 |
| #epoch | 1 | 1 | 1 | 1 | 1 | 1 | 1 | 1 |

(2020) for strong augmentations. Finally, the backbone of the classifiers is from this codebase: `https://github.com/bearpaw/pytorch-classification`.

**Table H.4:** Training details for CV experiments

| Parameter | Description | Value |
|---|---|---|
| $\tau$ | Pseudo-labeling confidence threshold | 0.95 |
| batch size | Number of labeled images per batch | 64 |
| $\mu$ | Ratio between number of unlabeled and labeled images in each batch | 7 |
| images per epoch | Number of labeled images per epoch | 65536 |
| #epoch | Number of epochs of training | 200 |
| lr | learning rate max value (10 epochs warmup then cosine decay) | 0.03 |
| weight decay | Weight decay regualrization coefficient | $5.00 \times 10^{-4}$ |
| momentum | Nesterov momentum for SGD optimizer | 0.90 |

## I  ADDITIONAL DETAILS

In Tables I.1, and I.2, we present some descriptive statistics of our CIFAR-10 synthetic image dataset to complement the samples shown in Figure C.1 and to help shed some light on the nature of the images generated by the NCSN network.

**Table I.1:** Unfiltered CIFAR-10 synthetic data statistics sorted by *Count*. The *Class* pseudo-label for each synthetic image is first obtained using a teacher model trained on the original CIFAR-10 data. *Count* denotes the number of images per class in the entire synthetic dataset (500K images). *Confidence* statistics shows the mean and standard deviation of the teacher model confidence score aggregated over each class.

| Class | Count | Confidence Mean | Std |
|---|---|---|---|
| truck | 64519 | 0.932 | 0.141 |
| ship | 32800 | 0.912 | 0.156 |
| horse | 39604 | 0.916 | 0.158 |
| frog | 76194 | 0.887 | 0.168 |
| dog | 38784 | 0.826 | 0.188 |
| deer | 38829 | 0.865 | 0.183 |
| cat | 65969 | 0.826 | 0.185 |
| bird | 37255 | 0.806 | 0.193 |
| car | 36264 | 0.936 | 0.140 |
| airplane | 69782 | 0.897 | 0.161 |

**Table I.2:** Filtered CIFAR-10 synthetic data statistics sorted by *Count*. The *Class* pseudo-label is first obtained for each synthetic image using a teacher model trained on the original CIFAR-10 data. The dataset is then filtered based on the teacher confidence score where only images with confidence $\geq \tau = 0.95$ are retained. *Count* denotes the number of images per class in the filtered synthetic dataset. *Confidence* statistics shows the mean and standard deviation of the teacher model confidence score aggregated over each class.

| Class | Count | Confidence Mean | Std |
|---|---|---|---|
| truck | 48796 | 0.996 | 0.009 |
| ship | 22741 | 0.995 | 0.010 |
| horse | 28498 | 0.996 | 0.010 |
| frog | 45923 | 0.993 | 0.012 |
| dog | 15984 | 0.989 | 0.014 |
| deer | 21413 | 0.993 | 0.012 |
| cat | 26311 | 0.988 | 0.014 |
| bird | 13440 | 0.988 | 0.014 |
| car | 28329 | 0.997 | 0.008 |
| airplane | 43745 | 0.992 | 0.012 |

