# OpenReview forum: "Generate, Annotate, and Learn: Generative Models Advance Self-Training and Knowledge Distillation"
_ICLR.cc/2022/Conference — ICLR 2022 Submitted_

### Official Review · Reviewer_uF4y · 2021-10-29

**Correctness:** 4
**Technical Novelty And Significance:** 3
**Empirical Novelty And Significance:** 3
**Recommendation:** 8
**Confidence:** 3

**Main Review:**

First, I want to thank the authors for the effort in writing this paper. It is extremely well-written and I appreciate the section 2 and 3, where key concepts are detailed and formally presented, very easy to follow. I wish more papers dedicated this much space and effort for adding context for the reader. I also appreciate the fact that you share situations where the models fail, showing limitations of the approach.

The model seems solid and justification for decisions made are clear. Improvements are not large but, given they are being compared to very strong baselines, it is fair to say they are relatively solid.

I do not have many weaknesses to highlight. Adding some of the things that could improve the publication below:

1 - For figure 1, a NIT comment is that adding the unlabeled data to the same level as the models made it a bit confusing at first glance. Maybe moving the "open domain unlabeled data" cloud more to the left might make it more clear where the whole process actually starts (maybe something similar for the "small amount of labeled data" part.
2- As you present the GLUE results on table 1, you split the scores with a "/". Since this is a paper that spans several areas of AI, it might be good to give more explanations on why this is the case and what are the metrics being considered in all your tables.
3 - For table 1 and 2, some standard deviation for the multiple runs or significance testing might be useful to compare results that are similar to each other.
4 - In section 5.3, a way to evaluate the impact of the quality for synthetic examples is to manually generate them by hand. I do think that is an important component for the paper that was not covered as well as the rest. Understanding the limitation of current generative models and understanding "how far we can go if we improve them" based on curated high quality data might make your point stronger.
5 - In table 5, you might want to add a baseline comparison without GAL so it is easier to compare the improvements.
6 - For tabular tasks, I think that section was more confusing than helpful. There was not a lot of analysis on why you decided to encode features the way you did and the impact it had on the two datasets where XGBoost outperforms your methods. It might be an interesting analysis to add or at least to discuss further.

**Summary Of The Paper:**

The paper provides a novel framework for advancing self-supervised learning, Knowledge distillation and few-shot learning for NLP and Tabular data. The framework focuses on self-training and knowledge distillation by generating synthetic data based on an unconditional generative model. They conduct extensive experiments spanning NLP, Tabular data and Computer vision datasets, discussing the benefits and limitations of their proposed methods. They also conduct and extensive literature review for each of the components used in their framework.

**Summary Of The Review:**

This work has extensive literature review and provides enough context for the reader to follow their thought process and the decisions made by the authors. Results are convincing and span multiple NLP tasks and research areas (tabular and CV data). There is in-depth discussions about limitations and benefits, and about the decisions made for the framework. My opinion is that this is a clear accept.

---

> ### Author Response · Authors · 2021-11-23
> **Response to Reviewer uF4y**
>
> We would like to appreciate the reviewer’s encouraging and valuable comments, and below is our detailed response.
>
>
> > Metrics used for different tasks
>
> Please refer to the general response
>
> > Standard deviation (or error bars) for tables:
>
> We would like to clarify that table 1 and table 3 are retrieved from GLUE leaderboard, which does not provide standard deviation. However for other tables, we provide the error bars below. The standard error values are calculated as: standard deviation of n independent runs divided by the square root of n. We also bold the values which are significantly better than the baseline (p<0.01)
>
>
> Error bar for Table 2:
>
> |Model                  | MNLI         | CoLA         | SST-2        | MRPC        | STS-B       | QQP       | QNLI             |  RTE          | Avg|
> | :-------- | :-------------- | :-------------- | :-------------- | :-------------- | :-------------- | :-------------- | :-------------- | :-------------- | :-------------- |
> |RoBERTa base   | 87.7$\pm0.1$ | 63.6 $\pm0.4$ | 94.8 $\pm0.1$ | 90.1 $\pm0.4$ |90.8 $\pm0.1$|  91.5 $\pm0.1$ |  92.6 $\pm0.1$ | 78.8 $\pm0.4$ | 86.2$\pm0.2$|
> |  + GAL (iter 3)     |87.9 $\pm0.1$ | **65.5** $\pm0.5$ | **95.3** $\pm0.1$ | **92.2** $\pm0.5$ | **91.7** $\pm0.2$|91.7 $\pm0.1$ | **93.2** $\pm0.1$  | **82.0** $\pm0.5$ | 87.4$\pm0.3$|
>
>
>
>
> Error bar for Table 7:
>
> | Model                |connect-4           |  Drug Review   |  Drybean           | Spambase|
> | --------                | --------------         |--------------         | --------------         |--------------         |
> | RoBERTa base | 85.0$\pm0.1$ | 84.6$\pm0.2$ | 85.0$\pm0.1$ | 87.7$\pm0.2$ |
> |   + GAL (iter 3)    | **87.3**$\pm0.2$ | **85.6**$\pm0.2$ | **85.9**$\pm0.1$ | **89.3**$\pm0.2$ |
>
>
> > Why did we use RoBERTa for the tabular datasets, given it underperforms XGBoost on two datasets?
>
> Connect4 and Drug Review consist of discrete and continuous features, while Drybean and Spambase consist of continuous features. To unify our framework and simplify the settings, we use RoBERTa as the backbone model, which is suitable for both discrete and continuous features. We will provide a discussion on this in the revised version.

---

### Official Review · Reviewer_Xyfj · 2021-11-02

**Correctness:** 3
**Technical Novelty And Significance:** 3
**Empirical Novelty And Significance:** 3
**Recommendation:** 5
**Confidence:** 4

**Main Review:**

The authors study how to handle the challenge of lacking task-specific unlabeled data in a semi-supervised learning (SSL). The problem is important and well-motivated.

Strengths:
*   The paper introduces a clear flow and comprehensive studies in the experimental section. The authors show many interesting findings. It is also surprising to see GPT-j could generate quality data with given few-shot labeled data.


Weakness:
*  First, the authors explicitly mention the small amount of labeled data in Figure 1.  Most of the experiments are conducted under full supervision except Table 4.  It is not very true that only a small amount of labeled data is available considering MNLI includes 400k examples and RTE also has more than thousands of labeled examples. This claim matters since data augmentation is much more challenging and important for low-resource scenarios.

* The proposed solution is to leverage data augmentation to generate task-specific unlabeled data. Considering that this idea is widely explored in the existing literature, I am more interested in the most distinguishing part: the authors use the unconditional autoregressive lanaguage model.  However, this claim is only investigated in Table 6. This still leaves many unanswered questions, like whether this observation is only validated for autoregressive models? especially considering that the previous papers[1] show class-conditional LM also brings sizable gains.

* Even though authors review many related works, they did not include other data augmentation works as baselines. Since this area is widely explored, it is important to understand how much benefits can be brought by this method contrasting other data augmentation options. The model with data augmentation and without augmentation is only ablation studies. Considering a GPT-2 is incorporated additional for data generation, it is not surprising to observe improvements. Comparison with other state-of-the-art data augmentation is needed. Besides the large language model as a data generator[1], many works like EDA[2], UDA[3] also explore this idea.

I have several additional questions:
1.  In table 1, what data is used for knowledge distillation in DistiRoBERTa + KD? Since KD is usually based on unlabeled data, I am not sure how the unlabeled data is achieved? If KD is based on trained labeled data, this seems not a fair comparison. This can only show that using additional LM for data augmentation is useful and such a claim is provided in many existing papers. I will suggest adopting other simple ways to generate unlabeled data like UDA[3] and still KD loss on them

2. I am not sure about the claim on class-conditional generative models. Since class-conditional generative models still generate noisy data, how about using pseudo labeling on class-conditional generated data? The negative effect seems to be brought by noisy labels instead of conditional generated data quality since codntional-class generation is also shown to be effective in a recent paper [4] with a more challenging setting.




[1] Jason W Wei and Kai Zou. 2019. Eda: Easy data augmentation techniques for boosting performance on text classification tasks

[2] Kumar, Varun  and Choudhary, Ashutosh  and Cho, Eunah Data Augmentation Using Pre-trained Transformer Models. worshop AACL 2020

[3] Qizhe Xie, Zihang Dai, Eduard Hovy, Minh-Thang Luong, Quoc V. Le Unsupervised Data Augmentation for Consistency Training. NeurIPS 2020

[4] Zirui Wang, Adams Wei Yu, Orhan Firat, Yuan Cao Zirui Wang, Adams Wei Yu, Orhan Firat, Yuan Cao


**Summary Of The Paper:**

The paper studies semi-supervised learning via generated synthetic data. More specifically, the authors handle the challenge of lacking task-specific unlabeled data and use unconditional language models to synthesize in-domain unlabeled data, showing the effectiveness of semi-supervised learning, knowledge distillation on NLP, and tabular tasks.

**Summary Of The Review:**

The authors study how to generate data for semi-supervised learning. The main claim is that the unconditional autoregressive language model is able to provide good quality data. However, this claim is still validated based on comparison with other state-of-the-art data augmentation methods.

---

> ### Author Response · Authors · 2021-11-23
> **Response to Reviewer Xyfj**
>
> We would like to appreciate the reviewer’s valuable comments and time, and below is our detailed response.
>
> > “Experiments are done under full supervision… It is not very true that only a small amount of labeled data is available”
>
> Sorry for the confusion caused by Figure 1. Please note that other than this part (we have updated Figure 1 accordingly), we did not claim that we’re working on a low-resource regime anywhere, except the few-shot settings. Instead, our major claim is that generating synthetic labeled data can advance the NLP tasks across different settings, including KD and few-shot learning.
>
> > Does our observation on unconditional autoregressive models fit in masked language models as class-conditional LM also brings sizable gains in [Kumar et al. 2020](https://arxiv.org/abs/2003.02245)?
>
> Thanks for the suggestion, which will be a very good direction for future work. We would like to remind the reviewer that Kumar et al. applied the class-conditional LM to a simulated scenario, where they subsample a small training set on each task. It has been shown that a low-resource regime benefits more from data-augmentation and self-training  ([Du et al. 2020](https://arxiv.org/abs/2010.02194), [Sohn et al. 2020](https://arxiv.org/abs/2001.07685)). However, our experiments show that the class-conditional LM fails in the full-sized data, which is consistent with [Ravuri et al. (2019)](https://arxiv.org/abs/1905.10887).
>
> To demonstrate this, we sample 100 instances from the synthetic RTE dataset generated by a class-conditional LM. Then we annotate these examples using a human annotator and the RoBERTa classifier. Finally, we compute the Accuracy, F1,  Precision and Recall scores between human labels and RoBERTa labels, and between human labels and conditioning labels (i.e., labels that the class-conditional LM conditions on):
>
> | label type         | Accuracy        | F1       | Precision       | Recall   |
> | :-------- | :-------------- | :-------------- | :-------------- | :-------------- |
> | RoBERTa  | 90.0  | 91.4  |  100.0 |  84.1 |
> | conditioning label | 72.0  | 71.4  |  66.0  | 77.8 |
>
> It is clear that the class-conditional LM has difficulty generating sentences retaining the semantics or pragmatics of a specified category, which also corroborates our theoretical analysis in section 4.2. On the other hand, RoBERTa is able to produce higher-quality labels that correlate better with human annotations.
>
> > Comparison with other state-of-the-art data augmentation.
>
> Please refer to the general response
>
> > “Since class-conditional generative models still generate noisy data.. using pseudo labeling on class-conditional LM”
>
> Our argument against class-conditional generative models in Section 4.2 under “How  about  class-conditional  generative  models?” only applies to the scenario that class-conditional generative models are used to synthesize *labeled* examples. That is, the conditioning label is used as the pseudo label. We have updated the text to make this clear. One can use class-conditional generative models to synthesize high-fidelity samples and successfully use such samples within GAL, as long as such samples are treated as unlabeled examples and annotated using a classifier.
> Following your suggestion, we added the results of using GAL on class-conditional generative models (with re-annotation) below:
>
> | Source of synthetic data         | SST-2        | RTE        | MRPC       | CoLA   |
> | :-------- | :-------------- | :-------------- | :-------------- | :-------------- |
> |No synthetic data (baseline) | 94.8 | 78.8 | 90.1 | 63.6|
> | Class-conditional LM  | 92.9 |   74.4 | 86.0 |  58.4 |
> | Class-conditional LM (GAL i.e., re-annotated)  | 95.4 | 81.0  | 91.4 |  65.2 |
> | Unconditional LM (GAL) | 95.3 | 81.4 | 91.7|  65.1|
>
> We observe that the difference between class-conditional LM (GAL) and unconditional LM (GAL) is negligible, but we suspect in certain problems, the use of class-conditional generative models (with re-annotation) will result in better performance, better the generative modeling problem can become easier when conditioning on rich labels.

---

> > ### Comment · Reviewer_Xyfj · 2021-11-29
> > **Thank you for your response**
> >
> > Thank you for your clarifications and new results to my questions.
> >
> > >  Our argument against class-conditional generative models in Section 4.2 under “How about class-conditional generative models?” only applies to the scenario that class-conditional generative models are used to synthesize labeled examples.
> >
> > Clarifications on this point change the argument proposed in the original paper a lot. Especially, the new statement about class-conditional generative models is not a new finding and is limited to a narrow scenario. Thus, it is difficult to claim the novelty of this paper relative to other data augmentation based on a language model. I will keep my original score.

---

> > > ### Author Response · Authors · 2021-12-01
> > > **Further discussion on class-conditional generative model (to Reviewer Xyfj)**
> > >
> > > Thank you for your valuable and constructive feedback.
> > >
> > > >Clarifications on this point change the argument proposed in the original paper a lot. Especially, the new statement about class-conditional generative models is not a new finding and is limited to a narrow scenario.
> > >
> > > The core argument of the original paper was that using class-conditional generative models to synthesize **labeled data**, as proposed by [Ravuri et al. (2019)](https://arxiv.org/abs/1905.10887) [Kumar et al. 2020](https://arxiv.org/abs/2003.02245), is suboptimal. Hence, the analysis focuses on this setting. The original submission did not discuss the use of class-conditional generative models to synthesize **unlabeled data**, followed by pseudo-labeling. We believe this is a subtle, but important point, as previous work does not consider the use of class-conditional generative models to synthesize **unlabeled data**. Given your comment, we have made this discussion clearer, and ran additional experiments that **clearly support** our analysis.
> > >
> > > We have difficulty understanding the reviewer’s comment about **limitation to a narrow scenario** as class-conditional generative models in the literature have been primarily used to synthesize **labeled data** [[Ravuri et al. (2019)](https://arxiv.org/abs/1905.10887) [Kumar et al. 2020](https://arxiv.org/abs/2003.02245)].
> > >
> > > >Thus, it is difficult to claim the novelty of this paper relative to other data augmentation based on a language model.
> > >
> > > We would appreciate it if the reviewer clarifies the other existing data augmentation techniques based on a language model that you are referring to.

---

### Official Review · Reviewer_zaeu · 2021-11-02

**Correctness:** 4
**Technical Novelty And Significance:** 3
**Empirical Novelty And Significance:** 4
**Recommendation:** 6
**Confidence:** 4

**Main Review:**

++ Clear & well written

++ relevant combination of different language models

++ strong & relevant experimental part

++ great last experiment on UCI tabular dataset

-- (probable) high computation cost

-- anonymization issue

The fact that GAL appears in the GLUE learderbord associated with the name XXX is a major issue if XXX is indeed the author of this article. As the GLUE leaderboard is mentioned a lot of time, this anonymization leak is important.

==

Few detailed comments:

Section 3 about distillation is very clear.

On Fig 1., authors could mention explicitely and/or graphically that GAL correspond to the bottom path (and not to the refining of the discriminative path).

Discussion of section 4.2 is clear and valuable.

Appendix are numerous and very interesting. They contain all information required to reproduced those experiments.

The fact that GAL appears in the GLUE learderbord associated with the name XXX [to be discussed privately] is a major issue if XXX is indeed the author of this article. As the GLUE leaderboard is mentioned a lot of time, this anonymization leak is important.

The authors do not discuss the computation cost of combining GPT2 & RoBERTa-large modeling.



**Summary Of The Paper:**

The authors propose a new framework of data augmentation based on generative models. The idea is to generate some new samples and then to classify them in an unsupervised manner to improve a student model. The authors provide an extreme experiment where the sentences are composed of numerical data raw extracted from UCI datasets. The authors give a very clear formalization & discussion to bridge between data generation and semi-supervised learning.
Then, the authors provide a large experimental section that show the interest of the GAL approach. In particular for small models but also for larger ones. The experimental section is very strong, investingating several interseting ablation. The last section (5.5) seems very promising for the future, in particular for data2text applications.

**Summary Of The Review:**

The proposed framework is simple, well described and very powerful. Experiments are numerous and relevant. I propose to accept this paper.

---

> ### Author Response · Authors · 2021-11-23
> **Response to Reviewer zaeu**
>
> We would like to appreciate the reviewer’s valuable feedback, and below is our detailed response.
>
> > Cost of the combination of GPT-2 and RoBERTa-large
>
> The fine-tuning of GPT-2 takes less than 30 minutes on average on GLUE datasets, while the fine-tuning of RoBERTa-large is less than two hours on average. The major overheads are the generation of 40x unlabeled synthetic data, which takes 12 hours on average. In total, we generate 38M usable sentences, while the retrieval approach ([Du et al. 2020](https://arxiv.org/abs/2010.02194)) needs to crawl more than 7B sentences as the datastore. Thus our approach is more effective and affordable. Please note that we will release the generated data to the community to encourage reproducibility and future study, upon acceptance.

---

### Official Review · Reviewer_Ke74 · 2021-11-02

**Correctness:** 4
**Technical Novelty And Significance:** 2
**Empirical Novelty And Significance:** 3
**Recommendation:** 5
**Confidence:** 4

**Main Review:**

## Strength

- Clearly written, Appendix also has extra descriptions of task and side experiments on image task and fine-tuning GPT2.
- Has good survey of related work in self-training.
- Extensive experiments on KD, self-training, and few-shot + tabular task.

## Weakness

The main challenge that this paper points out is in generating (or coming up with) the in-domain data.  Given this challenge,

- I am not entirely sure whether this GPT-2 fine-tuning or GPT3 generation is very novel for this paper to be accepted as ICLR paper.
- As *generation from fine-tuned large LM* is the main contribution of this paper, I suggest the authors to compare with other possible ways of collecting this in-domain data (such as kNN or generation without fine-tuning) to contrast and highlight their scientific contribution.


Thinking about other possible ways to generate in-domain data,
- This paper points out that it is difficult to collect in-domain data such as pair-wise sentences and tabular data is hard to get.
    - Q) For tasks that simply require single sentence as an input, what happens if you just use NYT, wikipedia corpus or if you do kNN given training set?
    - Q) For the pair-wise input (x1, x2), what happens if you simply do nearest-neighbor with x1, x2 given popular unlabeled texts?
    - Q) For the tabular dataset, I am not sure if pre-training of GPT2 really adds value, what happens if you just train separate LM for the tabular data.

**Other questions/comments**

- Have you considered using adapters instead of finetuning whole GPT2 which seems too expensive.
- What would be the result like if you don't really finetune GPT2 and do prompt-based generation? (not for few-shot)
- How about using g(x) notation in figure 1.
- Do you require such a large pre-trained model such as GPT2 for something like tabular dataset? I am not sure how GPT2 pretraining helps the generation of feature-like datasets such as tabular data.
- Maybe inappropriate to say outperforming XGBoost on 2 out of 4 tasks? (I think showing gains are fine over transformers, but besides connect-4 in Table 7, I am not sure the comparison to the XGBoost is that meaningful. The gaps between XGBoost and RoBERTa are remaining similar)
- Table 2, "We see consistent improvements with more GAL iterations,", I am not sure if MNLI and RTE show that?
- What are the two metrics (metric1/metric2 format) for MNLI, MRPC, STS-B, QQP? I recommend that the authors denote metrics for all their tables.
- Any reason that table 6 has part of GLUE only?
- How big is one iteration of GAL? Does it correspond to running through all 40x unlabeled data?

**Grammar**
If we had access to the oracle generative process, we were able to obtain the best KD and SSL results —> we could / we would be able to (?)

**Summary Of The Paper:**

This paper identifies challenges in self-training as a lack of in-domain data (input x). To overcome this challenge, the paper proposes to "generate" in-domain data using large self-supervised LMs. The rest of the process, which are annotation and learning, follows typical self-training and this paper takes the approach of learning with soft-target as (Du et al.)
The paper examines the proposed approach (GAL) on GLUE benchmark on KD (table1), SSL on full data (tab2,3), and prompt-based few-shot (tab4), and also examines GAL in tabular tasks (tab 7). On KD, GAL usually shows the largest improvement on 1st iteration and shows minor fluctuations of performance during more iterations (Tab2, 7).

**Summary Of The Review:**

I want to first thank the authors for writing a clear paper and conducting extensive experiments.

The experimental results show performance gains, however, I am not convinced that this approach is novel as it has been shown in previous work that self-training can improve performance and as we know that GPT-2, 3 can generate quality examples.

To appreciate the result more, I recommend authors to actually focus on discussing challenges in finetuning generator given training data. If they could improve the quality of generation in a novel way, maybe the paper could become more appealing. At the current state, I feel like GAL is very straightforward self-training with the unlabeled data generation with large-LM generation.
(Additional comments:
In case I missed some contributions of this paper, If we remove the good quality of GPT2,3  what are the benefits that GAL is bringing as a framework?  Could you elaborate on why generating in-domain data is a novel idea?)

---

> ### Author Response · Authors · 2021-11-23
> **Response to Reviewer Ke74**
>
> We would like to appreciate the reviewer’s valuable time and feedback, and below is our detailed response.
>
> >”I am not entirely sure….is very novel…Could you elaborate on why generating in-domain data is a novel idea?”
>
> We do not claim novelty for GPT-2 fine-tuning or text generation using GPT-3. Rather, the novelty lies in the application of language models to generate synthetic unlabeled text to help with knowledge distillation and few-shot learning. To our knowledge this is the first paper that shows improvements on KD and few-shot learning using synthetic text. We believe the simplicity of the idea is a key advantage, especially given that we are able to establish a new state of the art, outperforming much more complex KD techniques.
>
> As noted in the paper, the specific approach to generating synthetic data is important. [Ravuri et al. (2019)](https://arxiv.org/abs/1905.10887) has shown that class-conditional high-fidelity synthetic images do not translate to accuracy improvements in downstream classification tasks, which is also observed in our preliminary experiments. We explain why using class-conditional models to generate *labeled* synthetic examples is not good idea, and propose the use of a pretrained classifier to pseudo-label synthetic examples, which translates into accuracy improvements. As Reviewer uF4y pointed out, our study can help understanding the limitations of generative models as a source for synthetic data and more generally.
>
> > Comparison to other possible ways of collecting this in-domain data (such as kNN or generation without fine-tuning)
>
> We appreciate the suggestion. Following your advice, we have included a strong baseline based on round-trip translation. Please see the general response for the details. In addition, we note that the original knn augmentation paper ([Du et al. 2021](https://arxiv.org/abs/2010.02194)) did not include any NLP task with multi-sentence input schemes. We believe this hints that this technique may not be most effective in such scenarios. In a sense, using a parametric language model is similar to using a lookup table, but leverages the remarkable generalization of neural nets in the same way that neural language models outperform n-gram language models (even with smoothing techniques). Nevertheless, we are happy to include an ablation using kNN lookup if the reviewer finds it necessary.
>
> Unfortunately, generating sentences using a large LMs (e.g., GPT-3) without fine-tuning is much more expensive than fine-tuning a smaller LM like GPT-2. Specifically, [West et al. 2021](https://arxiv.org/abs/2110.07178) has shown that generating 6.4M datapoints costs 6K USD, whereas in total we have generated 38M usable sentences, which requires 36K USD. In fact, in our experiments, the cost of fine-tuning GPT-2 is negligible compared to generating 40x data.
>
> > Metrics used for GLUE
>
> Thank you. Please refer to the general response.
>
> > Using an adapter instead of fine-tuning whole GPT-2
>
> Thanks for the suggestion. According to our experiments, the fine-tuning of GPT-2 takes less than 30 minutes on average on GLUE datasets. The major overheads are the generation of 40x unlabeled synthetic data taking 12 hours on average, which cannot be sidestepped regardless of the use of the adapter. But we agree that exploring adaptors is an interesting direction for future work.
>
> > Besides connect4, the gaps between RoBERTa and XGBoost are still sizeable
>
> Our work aims to provide a universal framework for the use of generative models as a source of synthetic data for NLP and tabular tasks. To unify the problem settings, we reuse existing NLP models for tabular tasks. Albeit such a simplification, our experiments still demonstrate the effectiveness of the proposed approach, in contrast to the RoBERTa baseline. Given the success on RoBERTa, we are optimistic that our approach is applicable to deep learning models ( [Yoon et al. 2020](https://vanderschaar-lab.com/papers/NeurIPS2020_VIME.pdf), [Arick et al. 2019](https://arxiv.org/abs/1908.07442), [Somepalli et al. 2021](https://arxiv.org/abs/2106.01342)) specializing in tabular data and outperforming XGBoost.
>
> > Only 4 datasets in table 6
>
> The goal of Table 6 is to show the impact of model size on a representative subset of tasks. If needed, we will expand the study and include other tasks in the revised manuscript.
>
> > Is one iteration trained on 40x unlabeled data?
>
> Correct

---

> > ### Comment · Reviewer_Ke74 · 2021-11-29
> > **Thank you for your response, however, a major concern seems to be left unresolved.**
> >
> > Thank you for your clarifications on various questions that I have made:
> > metrics for GLUE benchmark, the definition of one iteration, and an additional experiment. However, with the comments below, I leave my score unchanged.
> >
> > **Major points**
> >
> > > Discussion about novelty.
> >
> > I agree that this paper does not claim the novelty of GPT-2 and that the focus is in the framework.
> > However, the framework of generate, annotate, and learn existed in the self-training community for a long time and this paper's contribution is on fine-tuning large LM such as GPT-2 (or prompting GPT-3).
> >
> > This is the part I am conflicted about the most. While the experiments are interesting and the writing is clear, the contribution/novelty is not too clear to me as the paper does not state the following points I mentioned in an earlier review:
> > "*If we remove the good quality of GPT2,3 what are the benefits that GAL is bringing as a framework? Could you elaborate on why generating in-domain data is a novel idea?**
> >
> > In general, the points in my initial review seem to be left unresolved even with the extra experiment on round-trip experiment.
> > > The experimental results show performance gains, however, I am not convinced that this approach is novel as it has been shown in previous work that self-training can improve performance and as we know that GPT-2, 3 can generate quality examples.
> >
> > >To appreciate the result more, I recommend authors to actually focus on discussing challenges in finetuning generator given training data. If they could improve the quality of generation in a novel way, maybe the paper could become more appealing. At the current state, I feel like GAL is very straightforward self-training with the unlabeled data generation with large-LM generation. (Additional comments: In case I missed some contributions of this paper If we remove the good quality of GPT2,3 what are the benefits that GAL is bringing as a framework? Could you elaborate on why generating in-domain data is a novel idea?)
> >
> > Currently, there are two experiments that can be used as comparisons to simple large-LM-finetuning. One is on RT and the other is class-conditional generation. However, as shown in the response to reviewer Xyfj, the class-conditional LM is focused on actually "annotation" step and if you annotate (or re-annotate), the performance of unconditional LM is very similar with class-conditional LM. As a result, there is only one experiment (RT vs. GAL) that really discusses "**different ways of generating**" which should be the main topic of the paper. I feel some more straightforward approaches (mabye kNN) should be compared to highlight the benefits of GAL.
> >
> > Moving to the next part, I question:
> > > What is the scope of GAL?
> >
> > With the comparison of the round-trip-translation (RT) experiment, I wondered why RT does not fall into the GAL framework and the short answer I find is again on "large-LM fine-tuning", i.e. RT is not GAL because it does not simply fine-tune large LM on a small set of in-domain data.
> >
> > But if this is the case then, I feel like why fine-tuning large LM (in combination with self-training) is such a novel idea has to be really discussed further. I do think there is a scientific value in the set of experiments that this paper presents, but **maybe the value is on *evaluating* the large LM's ability to generate good in-domain data** rather than the novelty on the framework of GAL?
> > In this sense, I agree with reviewer uF4y's point that *this study can help understand the limitations of genitive models as a source for synthetic data*. But in order to achieve such goal, maybe the paper's main focus should perhaps be on the above sentence marked with bold font?
> >
> > **Minor ponts**
> >
> > > we note that the original knn augmentation paper (Du et al. 2021) did not include any NLP task with multi-sentence input schemes. We believe this hints that this technique may not be most effective in such scenarios ...
> >
> > I am not too sure whether it actually "hints" such issues in effectiveness. Why not augment only one side of text for sentence A, B to A, B' where B' is in the k-nn of A?
> >
> >
> > > Description about XGBoost
> >
> > My comment on XGBoost was to point out inappropriateness of the key-contribution descriptions: "**GAL advance self-training for tabular tasks, outperforming XGBoost on 2 out of 4 tasks**." And yes, I understand that in a transformer-based setup that this helps as acknowledged in my first review. And again, I am not entirely sure why pertained LM is needed in these experiments.
> >
> > (my original review)
> > >Maybe inappropriate to say outperforming XGBoost on 2 out of 4 tasks? (I think showing gains are fine over transformers, but besides connect-4 in Table 7, I am not sure the comparison to the XGBoost is that meaningful. The gaps between XGBoost and RoBERTa are remaining similar)
> >
> > With above comments, I leave my score unchanged.

---

> > > ### Author Response · Authors · 2021-12-01
> > > **Further discussion on the major concern (to Reviewer Ke74)**
> > >
> > > Thank you for your detailed and constructive feedback.
> > >
> > > >While the experiments are interesting and the writing is clear, the contribution/novelty is not too clear to me as the paper does not state the following points I mentioned in an earlier review: "If we remove the good quality of GPT2,3 what are the benefits that GAL is bringing as a framework? Could you elaborate on why generating in-domain data is a novel idea?*
> > >
> > > We agree with the reviewer that no individual component of the proposed technique is new. The novelty of our work is not in LM fine-tuning or the pseudo labeling procedures, but rather in the novel way that they are combined -- that is pseudo labeling with **synthetically generated** data as opposed to **real** in-domain unlabeled data. The magnitude of the improvement we obtain over previous work indicates that our empirical findings are new to the research community. In addition, our **analysis** of class-conditional data augmentation [[Kumar et al. (2020)](https://arxiv.org/abs/2003.02245), [Ravuri et al. (2019)](https://arxiv.org/abs/1905.10887)] in Section 4 is novel and sheds light on why existing class-conditional techniques are not as effective as ours. This analysis is confirmed by the empirical results in Section 5.5 and Appendix F. We believe that our work will have a substantial impact, and all reviewers appear to agree as they find the experiments interesting.
> > >
> > > >Currently, there are two experiments that can be used as comparisons to simple large-LM-finetuning. One is on RT and the other is class-conditional generation. However, as shown in the response to reviewer Xyfj, the class-conditional LM is focused on actually "annotation" step and if you annotate (or re-annotate), the performance of unconditional LM is very similar with class-conditional LM. As a result, there is only one experiment (RT vs. GAL) that really discusses "different ways of generating" which should be the main topic of the paper. I feel some more straightforward approaches (maybe kNN) should be compared to highlight the benefits of GAL.
> > >
> > > The proposed GAL framework focuses on synthetic unlabeled data and is agnostic to the use of class-conditional or unconditional LMs as long as the generated examples are unlabeled. We explored two other ways of generating synthetic data: (1) using class-conditional generative models to synthesize labeled data, (2) using round-trip (RT) translation to synthesize unlabeled data. We agree that other methods including kNN can be applicable, but we believe the magnitude of the improvement we obtain over previous work and the two baselines above should be sufficient for acceptance at ICLR.
> > >
> > > >What is the scope of GAL? With the comparison of the round-trip-translation (RT) experiment, I wondered why RT does not fall into the GAL framework and the short answer I find is again on "large-LM fine-tuning", i.e. RT is not GAL because it does not simply fine-tune large LM on a small set of in-domain data.
> > > >But if this is the case then, I feel like why fine-tuning large LM (in combination with self-training) is such a novel idea has to be really discussed further. I do think there is a scientific value in the set of experiments that this paper presents, but maybe the value is on evaluating the large LM's ability to generate good in-domain data rather than the novelty on the framework of GAL? In this sense, I agree with reviewer uF4y's point that this study can help understand the limitations of genitive models as a source for synthetic data. But in order to achieve such goal, maybe the paper's main focus should perhaps be on the above sentence marked with bold font?
> > >
> > > Thank you. This is an important point. We agree that round-trip translation falls into the GAL framework as it provides another way to synthesize unlabeled examples. We need to discuss this better in the paper. The main claim of the paper is that the use of large LMs within the GAL framework is particularly effective for knowledge distillation and few-shot learning, as it advances the state of the art. We agree that framing the contribution of the paper around “evaluating the large LM's ability to generate good in-domain **unlabeled** data” is more appropriate. We will reframe our key contribution as studying the use of large LMs within GAL in the final version of the paper if accepted. In addition, we believe the discussion and analysis of synthetic **unlabeled** data is central to the paper.
> > >
> > > >Minor points
> > >
> > > We agree that kNN can be applied to each sentence in a sentence pair separately. We did not think of this. We will try our best to include this ablation in the final version of the paper. We agree that XGBoost is not directly comparable and we will soften the language around improvement over XGBoost.

---

### Author Response · Authors · 2021-11-23
**General response**

Dear reviewers,

Thank you for your valuable feedback. We have incorporated your feedback and made some improvements to the paper, which are marked in blue color in the paper.

> Review Ke74 and Xyfj: Comparison to other possible ways of collecting this in-domain data

Following your suggestion, We compare with round-trip translation (RT), a strong data-augmentation baseline (e.g.,[Yu et al., 2018](https://arxiv.org/abs/1804.09541); [Shleifer, 2019](https://arxiv.org/abs/1903.09244)).   We mirror the experimental setup of GAL and generate 40x unlabeled data using German as the bridge language (English>German>English). The translations are generated by the best model in WMT19 (Ng et al., 2019; [fairseq](https://github.com/pytorch/fairseq/tree/main/examples/translation) ). Although DistilRoBERTa+RT is better than vanilla DistilRoBERTa and KD variants, it still significantly underperforms our approach. Here are the results:

|Model                  | MNLI         | CoLA         | SST-2        | MRPC        | STS-B       | QQP       | QNLI             |  RTE         | Avg|
| :-------- | :-------------- | :-------------- | :-------------- | :-------------- | :-------------- | :-------------- | :-------------- | :-------------- | :-------------- |
DistilRoBERTa |  83.8/83.4	| 55.9 | 93.2 | 87.4/83.1 | 87.5/87.5 | 71.7/89.1 | 90.6 | 73.3 |  81.2|
DistilRoBERTa+RT | 86.1/86.1| 53.0 | 94.6  | 91.0/87.8 |	89.2/88.8 | 73.1/89.9 | 	92.4 |	 76.9 | 82.7 |
DistilRoBERTa+gal | **87.4/86.5** | **60.0**	| **95.3** | 	**91.9/89.2** |	**90.0/89.6**	| **73.3/90.0** | 	**92.7** |	 **81.8** | **84.8**|

We have updated Table 1 of the paper and included the DistilRoBERTa+RT baseline, which highlights the strength of GAL.

> Review Ke74 and uF4y: Metrics used for GLUE

We provide the metrics used for GLUE benchmark below:

| CoLA         | SST-2        | MRPC        | STS-B       | QQP       | QNLI             |  RTE         |
| :-------- | :-------------- | :-------------- | :-------------- | :-------------- | :-------------- | :-------------- |
|Matthew's Corr | Accuracy | F1 / Accuracy | Pearson/Spearman Corr|F1 / Accuracy | Accuracy| Accuracy|

MNLI is evaluated on two sets: MNLI matched and MNLI mismatched. Both of them use accuracy. We report accuracy for tabular datasets and few-shot settings. Following your question, we clarified the used metrics in the captions of Table 1.

---

### Decision · Program_Chairs · 2022-01-20

**Decision:**

Reject

**Comment:**

To overcome the challenge of lacking task-specific unlabeled data in semi-supervised learning (SSL) or knowledge-distillation (KD) tasks, this paper presents a new framework called "generate, annotate, and learn (GAL)" that uses unconditional language models to synthesize in-domain unlabeled data to advance SSL and KD. Extensive experiments on both NLP and tabular tasks demonstrate positive results of the proposed method.

Reviewers generally agree on several key strengths of the paper, e.g., the paper is well-written, literature review is comprehensive, experimental results are generally positive (the improvements over the standard baselines on GLUE benchmark looks solid despite not very significant). On the negative side, some reviewers did raise some major concerns about the novelty of the proposed framework and the lack of strong baselines for comparison. For example, the proposed GAL framework doesn’t seem particularly novel as neither of the proposed components is new, and the key value of the work seems on the contribution of evaluating the large LM's ability to generate good in-domain unlabeled data (as agreed by authors). Therefore, it is very important to compare with other existing data augmentation baselines in the empirical studies. While authors did try to add one round-trip-translation (RT) data augmentation baseline for comparison during the rebuttal, more stronger SOTA data augmentation baselines should be compared.

Overall, this is a good paper which is worthy of publication in near future but it still needs some more work on more extensive comparison of more baselines and improvements on the writing of novelty and contribution claims.